



**Design and evaluation of BOOGIE: a collector for the analysis**
**of cloud composition and processes: Biological, Organics,**
**Oxidants, soluble Gases, inorganic Ions and metal Elements**
Mickael Vaitilingom[1,2*], Christophe Bernard[3], Mickaël Ribeiro[1,3], Christophe Berthod[4],
Angelica Bianco[1], Laurent Deguillaume[1,3*]
[1] Laboratoire de Météorologie Physique, UMR 6016, CNRS, Université Clermont Auvergne, 63178 Aubière,
France
[2] Laboratoire de Recherche en Géosciences et Énergies, EA 4539, Université des Antilles, 97110 Pointe-à-Pitre,
France
[3] Observatoire de Physique du Globe de Clermont-Ferrand, UAR 833, CNRS, Université Clermont Auvergne,
63178 Aubière, France
[4] Division Technique de l'Institut National des Sciences de l'Univers, UAR 855, CNRS, 91190 Gif-sur-Yvette,
France.
*Correspondence to:* Laurent Deguillaume (laurent.deguillaume@uca.fr) and Mickael Vaitilingom
(mickael.vaitilingom@univ-antilles.fr)
**Abstract.** Cloud/fog droplets comprise a myriad of chemical compounds and are living environments in which
microorganisms are present and active. These chemical and biological elements can evolve in various ways within
the cloud system, and the aqueous transformation of chemicals contributes to atmospheric chemistry. In situ cloud
studies are fundamental in this sense, because they enable us to study the variability in cloud chemical composition
as a function of environmental conditions and assess their potential for transforming chemical compounds. To
achieve this objective, cloud water collectors have been developed in recent decades to recover water from clouds
and fogs using different designs and collection methods. In this study, a new active ground-based cloud collector
was developed and tested for sampling cloud water to assess the cloud microbiology and chemistry. This new
instrument, BOOGIE, is an easy mobile sampler for cloud water collection with the objective of being cleanable
and sterilisable, respecting chemical and microbial cloud integrity, and presenting an efficient collection rate of
cloud water. Computational fluid dynamics simulations were performed to theoretically assess the capture of cloud
droplets by this new sampler. Few turbulences have been observed inside the collector and a 50% collection
efficiency cutoff of 10 µm has been estimated. The collector was deployed at Puy de Dôme station under cloudy
conditions for evaluation. The water collection rates were measured at $156 \pm 52$ mL h$^{-1}$ for a collection of 17 cloud
events; considering the measured liquid water content, the sampling efficiency of this new collector has been
estimated at $87.2 \pm 8.6\%$ over the same set of cloud events. BOOGIE was compared with other active cloud
collectors commonly used by the scientific community (Cloud Water Sampler and Caltech Active Strand Cloud
Collector version 2). Four cloud events were collected; the three samplers presented similar collection efficiencies
(between 79% and 88% on average). The measured ionic composition was comparable even if differences were
highlighted between collectors, the consequence of different designs, and the intrinsic homogeneity in the chemical
composition within the cloud system.
**Keywords:** Cloud chemistry, monitoring, cloud water collector, chemical composition, biological composition.



## 1 Introduction

The chemical composition of clouds is highly complex because it results from various processes: (1) the mass transfer of soluble compounds from the gas phase into cloud droplets, (2) dissolution of the cloud condensation nuclei released into the aqueous phase as a complex mixture of soluble molecules, and (3) photochemical and biological transformations leading to new chemical products (Herrmann et al., 2015).

Field experiments to characterise this multiphasic medium were developed in the 1950s but increased in the 1980s because of precipitation acidification through sulphur oxidation in cloud droplets (Munger et al., 1983; Hoffmann, 1986; Kagawa et al., 2021). These studies have highlighted that cloud and fog processing is efficient and plays a major role in air pollution by transforming gases and aerosol particles. Numerous investigations have focused on inorganic compounds that control aqueous-phase acidity (Pye et al., 2020). The production of strong acids has been assessed because it increases particle mass when clouds/fogs evaporate and leads to acidic deposition when clouds precipitate (Tilgner et al., 2021). Early in the 1990s and much more so in the 2000s, researchers investigated the composition of dissolved organic matter in cloud/fog water which has multiple natural and anthropogenic sources of primary or secondary origins (Herckes et al., 2013). Based on scientific issues, specific classes of compounds have been targeted, such as short-chain carboxylic acids and carbonyls (Löflund et al., 2002; Munger et al., 1995; Sun et al., 2016) and more recently carbohydrates and amino acids (Triesch et al., 2021; Renard et al., 2022). Attention has also been paid to the detection of pollutants with strong sanitary effects, such as HAP, phenols, and phthalates (Lüttke et al., 1999; Li et al., 2010; Lebedev et al., 2018; Ehrenhauser et al., 2012) because they can impact ecosystems through precipitation (Wright et al., 2018). Recent investigations using high-resolution mass spectrometry have revealed the complexity of the organic matrix, with thousands of detected molecules (Zhao et al., 2013; Cook et al., 2017; Bianco et al., 2018; Sun et al., 2021). This organic matter is processed during the cloud lifetime and has raised new scientific questions such as the formation of secondary organic aerosol by aqueous phase reactivity ("aqSOA") (Blando and Turpin, 2000; Lamkaddam et al., 2021) and light absorbing material referring to brown carbon ("BrC") (Laskin et al., 2015). Microorganisms are also present and active in cloud droplets (Amato et al., 2005; Vaïtilingom et al., 2012; Xu et al., 2017; Hu et al., 2018). They can be incorporated because they serve as cloud condensation nuclei (Bauer et al., 2002; Deguillaume et al., 2008) and can impact cloud water composition through their metabolism by consuming or producing new molecules (Liu et al., 2023; Vaïtilingom et al., 2013; Pailler et al., 2023). Many investigations have focused on biological cloud characterisation (Amato et al., 2017; Wei et al., 2017).

Monitoring cloud chemical and biological compositions is crucial for evaluating the role of key environmental parameters such as emission sources, atmospheric transport and transformations, and physicochemical cloud properties such as cloud acidity or microphysical cloud properties (liquid water content [LWC] and size distribution of cloud droplets). Specific sites or aircraft campaigns allow the collection of cloud water influenced by marine (Macdonald et al., 2018; Gioda et al., 2011), continental (Van pinxteren et al., 2016; Hutchings et al., 2009; Lawrence et al., 2023; Van Pinxteren et al., 2014) and urban emissions (Li et al., 2020; Guo et al., 2012; Herckes et al., 2002) over various continents (mainly Europe, North America, Asia). Owing to their poor accessibility and remoteness, certain geographical locations have been less investigated, such as the Arctic region (Adachi et al., 2022), tropical environments (Dominutti et al., 2022), or marine surfaces (Van Pinxteren et al., 2020). Field experiments combining cloud water and gaseous phase chemical characterisation have also been



conducted to evaluate the partitioning of molecules between these two phases and whether bulk cloud water obeys
Henry's law (Van Pinxteren et al., 2005; Wang et al., 2020). Bulk aqueous cloud media are used for laboratory
investigations to study the aqueous transformations induced by light and the presence of microorganisms
(Schurman et al., 2018; Bianco et al., 2019).
Therefore, the scientific community requires regular and long-term measurements of cloud chemical and biological
parameters. However, cloud sampling procedures are challenging. In recent decades, different samplers have been
developed and deployed in the field, which can be operated under specific environmental conditions and present
different collection efficiencies possibly impacted by meteorological conditions. These are commonly based on
the impact of cloud droplets on the collector surface and avoid the collection of small droplets ($<5$ µm in diameter).
Their collection efficiency and 50% collection cutoff diameter (d50) were calculated and estimated to evaluate the
accuracy of droplet collection by the sampler. Monitoring of the microphysical cloud properties (LWC and size
distribution) is required to assess this. These samplers refer to "bulk" cloud water collectors because they group
droplets of different sizes. Many types of collectors can be listed: active or passive ground- or aircraft-based, and
single- or multi-stage. Passive collectors are dependent on wind speed because the air needs to flow through them,
allowing sampling. Active collectors are ground-based collectors through which air-containing droplets are forced
to flow inside the system by devices such as pumps or ventilator fans. They have been designed and commonly
used to obtain higher volumes of water required for laboratory investigations. Ground-based samplers are easy to
install, inexpensive, and suitable for long-term observations. Samplers installed on aircrafts are less widely used,
and recent developments by Crosbie et al. presenting a new axial cyclone cloud water collector have shown to
strongly improve the collection efficiency of cloud droplets compared to previous samplers (Crosbie et al., 2018).
All these samplers are described in reviews where and their designs, their advantages, and limitations are presented
(Roman et al., 2013; Skarżyńska et al., 2006).
Two types of ground-based active samplers are often used by the scientific community to monitor cloud chemistry
and microbiology: the Cloud Water Sampler (CWS) from Vienna University (Kruisz et al., 1993) and the Caltech
Active Strand Cloudwater Collector (CASCC) from Caltech University (Daube et al., 1987; Demoz et al., 1996;
Collett Jr et al., 1990). These collectors have been adapted for long-term monitoring (Gioda et al., 2013; Guo et
al., 2012; Deguillaume et al., 2014; Renard et al., 2020) and specific field campaigns (Wieprecht et al., 2005;
Van pinxteren et al., 2016; Li et al., 2017; Li et al., 2020; Bauer et al., 2002).
The Puy de Dôme (PUY) station is a reference site for the collection of cloud water from samples collected between
2001 and the present. The sampler was obtained from Kruisz et al. (1993) and has been widely used for microbial
and chemical atmospheric studies at this site (Marinoni et al., 2004; Marinoni et al., 2011; Bianco et al., 2017; Joly
et al., 2014) This model can collect wet or supercooled droplets, even at high wind speeds. It is made of aluminium
or Teflon; the collection vessel can be removed for sterilisation and cleaning. However, the collected water volume
of 10–60 mL per hour is a limit for chemical and microbial analyses that require increasing volumes. For long
collection times, the vessel should be removed regularly to transfer the water into a sterile storage bottle. These
manipulations expose the samples to contamination. The aspiration system must be powerful and, consequently,
heavy and energy-consuming, which limits mobile sampling. The objective of this study was to present a new
ground-based cloud collector that responds to different constraints. This tool should be suitable for analysing cloud
microbiology and chemistry, easy to clean and sterilise, allow the collection of high volumes of water, and be easy



to deploy for field campaigns (light and low energy consumption). To achieve these objectives, we developed a
BOOGIE collector. This study describes this instrument and compares it to other commonly used samplers to
evaluate its efficiency.

## 2 Materials and Methods

### 2.1 Conception of the BOOGIE cloud collector

The 3D drawing was performed with Autodesk® Inventor 2016 and recently updated using the 2019 version. The
prototype of the collector used in this study was fabricated on an aluminium stand (Al 5754 and 6060). This
material exhibits robust properties and can be easily sterilised by autoclaving before field collection. Aluminium
plates were cut using a laser and folded using a metal press. The collection funnel was adapted to a GL 45 thread
to directly screw borosilicate glass or polytetrafluoroethylene (PTFE) bottles. All the aluminium parts were treated
by QUANALOD® anodisation, with thickness of 20 µm, suitable for aluminium objects exposed to harsh
environmental conditions. All parts were thoroughly cleaned to eliminate all manufacturing residue and several
cycles of sterilisation by autoclaving (121°, 20 min per cycle) were performed to clean the collector.
The vacuum inside the collector was ensured by an axial fan (EMB-papst©, model 6300TD, S-Force, 40 W, 12 V
DC) able to work under wet conditions and temperatures of -20 °C to 70 °C. It has a fan diameter of 172 mm and
a maximum flow capacity of 600 $m^3$ $h^{-1}$ (manufacturer data). It is equipped with a controlled voltage for speed
setting, which allows modulation of the fan velocity according to 10 increasing intensities. To measure the air inlet
and outlet velocity, a thermal anemometer efficient from 0.2 to 20 m $s^{-1}$ was used (model Lutron AM-4204 from
RS PRO©).

### 2.2. Computational Fluid dynamics (CFD) simulations

Finite element modelling and simulations were performed using Simcenter 3D software from Siemens Industry
Software Inc., version 2022.1. The solver environment was Simcenter 3D Thermal/Flow Advanced Flow. The
flow and particle tracking solvers are proprietary to Maya Heat Transfer Technologies. Other numerical
computations and figures were performed using MATLAB version 2021a.
The fluid domain is represented by the inner volume of the collector. To compute a realistic flow inside the
collector, it is necessary to consider the structure of the collector, which is composed of thin walls and metal plates,
to enable air deflection and the collection of cloud water droplets. The Simcenter 3D software allows the generation
of a volume or mesh directly from the boundaries of different parts of the collector; however, this method was
unsuitable because of the thin inner walls. The fluid domain was built using successive Boolean subtractions by
leaving a void in the right place, leading to a realistic geometry of the air volume (**Figure S1a**).
A finite element mesh was created using CTETRA4 solid elements. The element size was variable: the internal
mesh size was set to 20 mm, whereas the element size was set to 24 mm on the rear faces next to the fan and to
only 4 mm on the front face, allowing air deflection and the collection of droplets (**Figure S1b**). The total numbers
of elements and nodes were 869 799 and 178 610, respectively.
For the air inlet flow, three slots of the collector front face were defined as the inlet flow boundary conditions. The
flow direction was perpendicular to the front face and the external absolute pressure was equal to the ambient





pressure. For the air outlet flow, air velocity was applied to the rear circular face representing the fan. The
magnitude varied according to the velocity ranges. The vector was perpendicular to the face.
The fluid is the standard air at the altitude of 1500 m (*i.e.*, summit of the PUY), at 15 °C, with the following
physical characteristics: 1.1 kg m$^{-3}$ for the mass density and 1.75 kg m$^{-1}$ s$^{-1}$ for the dynamic viscosity.
The outlet velocity of the fan can be modulated among 10 intensities. The resulting air inlet volume flows have
been measured using a hot-wire anemometer located in front of the slots. The surface area of the fan outlet was
17671 mm$^2$, and the total area of the three inlet slots was 11088 mm$^2$. Therefore, there was a theoretical ratio of
1.6 between the air inlet volume flow and the air outlet volume flows. To agree with the measured air inlet volume
flow, the outlet velocities for the collector simulations were varied for the CFD simulations between 1 and 10 m
s$^{-1}$ in 1 m s$^{-1}$ step.
Different particles were used in the simulation. The water drops were injected into the flow at the three air-inlet
slots. Eight different values of drop diameter were selected between 5 and 20 μm. The water droplets were
considered spherical. The drag coefficient was automatically calculated using the Reynolds number. The density
of water was assumed to be 1 kg/dm$^3$. Gravity was applied to the cloud particles, and the gravity vector was defined
as the –Z axis with an acceleration amplitude of 9.81 m s$^{-2}$. The sizes and masses of each particle class are
summarised in **Table S1**.
In the air flow inside the collector, three vertical plates participated in droplet collection. If cloud water drops
impact them, they should flow to the bottom of the funnel. Therefore, there is a specific surface configuration; if
the water drops stick to the collection face, they do not rebound.
We selected the fully coupled pressure-velocity solver to solve the mass and momentum equations simultaneously
for each time step. The solver iterates the pressure and velocity solutions until convergence is achieved at each
time step. Modelling fluid flow turbulence is crucial for accurately simulating airflow. The flow solver uses
different turbulence models that add a viscosity term to the Navier–Stokes governing equations. The two-equation
model computes the viscosity term using two additional equations that are solved in parallel with the Navier–
Stokes equations. Among the two-equation models, the k-omega model was selected for this study. The steady
state time step was fixed to 0.01 s for all the model simulations.
For the steady-state simulation, the flow was fully developed, and its properties (velocity, pressure, and turbulence)
were used in the particle-tracking equation. During the analysis, the software solved the equation of motion for
each particle once per time step. Notably, because the particle tracking simulation is independent of the flow
simulation, the particles do not affect the 3D flow. The injection duration in the fluid domain was 60 s, which is a
good compromise between the relevant calculation and a reasonable simulation time.
**2.3 Experiments: inter-comparison of samplers**
**2.3.1 Sampling site**
The testing site of the different cloud collectors was the observatory of the PUY summit at 1465 m above sea level.
It is part of the C̲ézeaux-A̲ulnat-O̲pme-P̲uy D̲e D̲ôme (CO-PDD) instrument platform for atmospheric research
(Baray et al., 2020). PUY is recognised as a global station in the Global Atmosphere Watch (GAW) network and
is part of the European and national research infrastructures Aerosol Cloud and Trace Gases Research





Infrastructure (ACTRIS) and the Integrated Carbon Observing System (ICOS). The PUY is often located in the
free troposphere, particularly during cloud events, and the characterised air is representative of synoptic-scale
atmospheric composition. Various biological, physical, chemical, and cloud microphysical parameters were
monitored on-site. For cloud microphysical properties, we use a ground-based scattering laser spectrophotometer
for cloud droplet volume measurements from Gerber Scientific, Inc. (Reston, VA, USA). All cloud microbiology
and    chemistry    data    are    available    in    the    PUYCLOUD    database    (https://www.opgc.fr/data-
center/public/data/puycloud).

### 2.3.2 Cloud collectors

Two bulk cloud collectors were compared with a newly developed BOOGIE collector. These are active ground-
based collectors commonly used in cloud field studies. They have different collection efficiencies, resulting in
different volumes of cloud water that can be sampled. Cloud water collectors are generally designed to avoid the
particles below 5 microns to avoid sampling the interstitial aerosol around the droplets. This is a compromise to
obtain a sufficient volume of water with less contamination from dry and deliquescent particles. Typically, the
smallest droplets were not sampled. The 50% collection efficiency cutoff, based on the droplet diameter, is often
predicted from the impaction theory and strongly depends on the aerodynamic design of the impactor and the air
flow rate (Berner, 1988; Schell et al., 1992). The collection efficiency for in situ conditions will depend on the
LWC, and the meteorological conditions could strongly perturb the way the collectors are able to impact cloud
droplets.

### *Caltech Active Strand Cloud water Collector: CASCC2*

A compact version of the original CASCC collector was used and lent by the Institut de Radioprotection et de
Sûreté Nucléaire (IRSN). This sampler, named CASCC2, was constructed according to the recommendations of
Demoz et al. (1996). It has an estimated cutoff diameter of 3.5 µm (droplet diameter collected with 50% collection
efficiency). The airflow passed through a set of six rows of stainless-steel strings (diameter, 0.5 mm) with a
velocity of 8.6 m s$^{-1}$. The strings were vertically tilted 35°. The collector design has been shown to generate a
stable airflow inside of 348 m$^3$ h$^{-1}$. The fraction of air sampled was calculated to be 86% (*i.e.*, 299 m$^3$ h$^{-1}$). The
volume fraction of the ambient droplet distribution collected was evaluated in Demoz et al. (1996), who showed
that this fraction is close to one over most of the LWC range (superior to 95% >0.1 g m$^{-3}$ of LWC). Therefore, we
can estimate at the end a resulting sampled airflow at 284 m$^3$ h$^{-1}$ (4.73 m$^3$ min$^{-1}$). Cloud droplets coalesce on the
strands and fall into a bottle through a Teflon tube owing to the combination of gravity and aerodynamic resistance.
A description of the sampler is provided in **Figure S2**.
The collector body was stainless steel, the inlet contained the impaction rows, and the sample drainage was
removed before each sampling for cleaning and sterilisation. A sterilised amber glass bottle was placed under the
sample drainage during collection. This cloud collector was not adapted for temperatures <0 °C because droplets
freeze upon impaction on metallic strains.

### *Cloud Water Sampler: CWS*

This collector (**Figure S3**) was developed specifically to collect warm and supercooled clouds, which can either
freeze upon impaction or be collected directly in the liquid phase (Kruisz et al., 1993; Brantner et al., 1994). It was
designed to sample cloud water for specific studies on the detection for example of fungal spores and bacteria in
cloud water (Tenberken-Pötzsch et al., 2000; Bauer et al., 2002). It comprises a single-stage impactor backed by a
large wind shield (50 cm wide and 50 cm high) installed in front of the wind. The wind velocities were reduced in
front of the shield, and the flow was directed into the single-slit nozzle. Cloud droplets ranging up to 100 μm in
diameter were estimated to be stopped in front of the shield, stay airborne, and were sampled from a stagnant flow.
Cloud droplets, which were drawn through a slit 25 cm long and 1.5 cm wide, collided on a rectangular aluminium
collection plate installed horizontally, and water was collected in a reservoir below the plate. This sampler model
presents an estimated cutoff diameter at 50% collection efficiency of 7 μm at a sampling rate of 86 m$^3$ h$^{-1}$, as
indicated in Brantner et al. (1994). The CWS used at the PUY was a homemade collector following the
recommendation formulated by Kruisz et al. (1993); however, the suction system presented its own characteristics,
with an inlet air velocity of 13.5 m s$^{-1}$.
The blower was placed under the sampler and connected to the collector body via tubing. This was built of
aluminium, and the collection plate and vessel were removable for cleaning and sterilisation. In contrast to the
CASCC2, in which the water sample flowed into a glass bottle, in the CWS, the water remained in the collection
vessel during the sampling period. It is not possible to check the collected volume during sampling, and the water
must be regularly removed by opening the collector and transferring it to a storage bottle. This collector has been
used for studies at PUY since the 2000s (Marinoni et al., 2004) because the collection plate and vessel can be
sterilised in the laboratory, allowing for microbial analysis of cloud waters.
**2.3.4 Chemical and microbial analysis**
Chemical and biological analyses were performed on the cloud samples following the standardised procedures
described in Deguillaume et al. (2014). The main ions (Cl$^-$, NO$_3^-$, NH$_4^+$, SO$_4^{2-}$, Na$^+$, Ca$^+$, Mg$^+$, K$^+$) were analysed
using ion chromatography. Formaldehyde and hydrogen peroxide levels were measured using derivatisation
methods and analysed by fluorimetry. Total microbial cell counts, including bacterial, yeast, and fungal spores,
were determined using flow cytometry. The microbial energetic state was determined by measuring ATP and ADP
concentrations using bioluminescence. More information has been added to this analysis in the Supplementary
Information.
**2.3.5 Back-trajectory analysis**
The CAT model (Baray et al., 2020) was used to estimate the air mass history reaching the summit of the PUY
Mountain during the cloud-sampling period. This model uses the ECMWF ERA-5 wind fields and integrates a
topography matrix; back trajectories were calculated every hour during cloud sampling; the temporal resolution
was 15 min, and the total duration was 72 h. These calculations are fully described by Renard et al. (2020).
**3 Results**
**3.1. Conception and operating principles of the BOOGIE collector**
The new collector is a single-stage collector that uses impaction to sample the cloud droplets (Marple and Willeke,
1976). The collector is designed as a slit impactor. **Figure 1** shows the assembled collector (left) and the different
parts of the collector and how they should be assembled for sampling. A GIF animation (**Movie 1**) showing the
assembly of the collector before sampling is provided in the Supplementary Information. A photograph of the



collector is shown in **Figure S4**, and all the dimensions are detailed in **Figure S5**. Parts 1, 2, and 5 were sterilised
by autoclaving before sampling to allow for biological analysis.

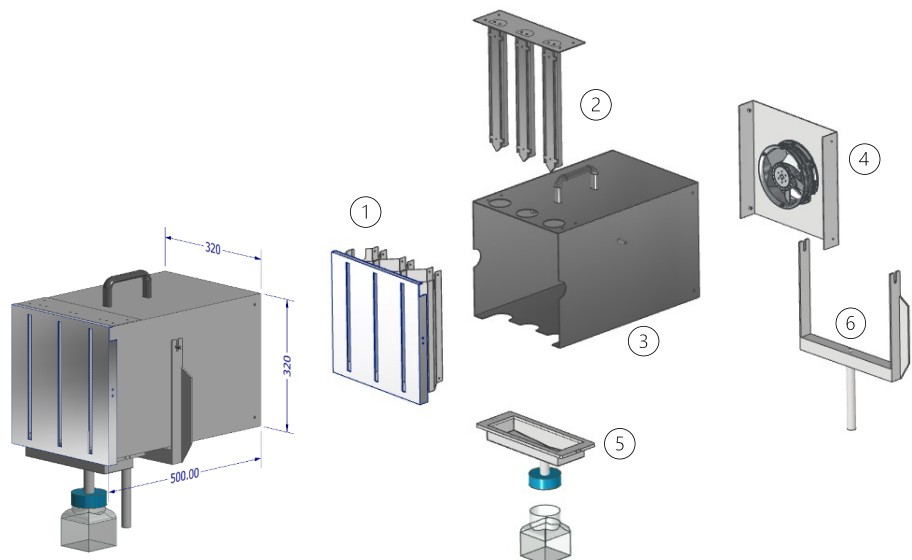


**Figure 1. Schematic of the design of the BOOGIE collector. Assembly of the different parts of the BOOGIE collector:**
**(1) front face with the three slots; (2) impaction plates; (3) collector body; (4) rear face with the fan; (5) funnel; (6)**
**instrument holder.**
The cloudy air entered via three rectangular inlets oriented vertically side by side, each 30 cm long and 1.2 cm
wide, with 9 cm between them. The droplets were impacted by inertia on aluminium plates located 45 mm behind
the air inlets. The inlet width and distance between the inlet and impaction plate were selected to be identical to
those of the CWS. The air and smaller noncollected droplets were directed to a shared corridor before the air fan.
The collected water flowed to the collection funnel under gravity, and the collection bottle was sterilised.
The fan can be modulated at 10 intensities (10–100% of the maximum fan speed). The air inlet velocities were
measured in front of each of the three slots of the BOOGIE collector at different heights (high, middle, and low
points), with the velocity modulated according to these 10 values (**Figure S6**). The measured velocities varied
from 2 to approximately 15 m s$^{-1}$, with an increase of approximately 1.5 m s$^{-1}$ per intensity step. The air inlet
velocity stabilised at 90% of the fan speed (corresponding to a value of 14 m s$^{-1}$). The velocities measured at
different fan intensities were highly homogeneous between slots and for the same slot at different heights, with
only a few percentages of standard deviations (between 1.5 to 5%), possibly indicating that the geometry of the
collector provided good airflow stabilisation. The next section, in which the flow inside the collector is simulated,
provides a more robust assessment of this statement.
**3.2 Performance evaluation**
**3.2.1 CFD simulations**
***Flow velocity***

292 **Figure 2a and b** displays the flow velocity field inside the collector for air outlet flow velocity equal to 8 m s$^{-1}$

293 (the same for 2 m s$^{-1}$ in **Figure S7a and b**). As noted in Section 2.2.2, the air outlet flow velocity equal to 8 m s$^{-1}$

294 corresponds to an air inlet flow velocity equal to 12.8 m s$^{-1}$ (1.6 factor), a value similar to the measured air inlet

295 velocity of the collector. We present the horizontal cutting planes at the centre of the fan. Regardless of the air

296 outlet velocity, the colour display of the flow velocity contour is identical.

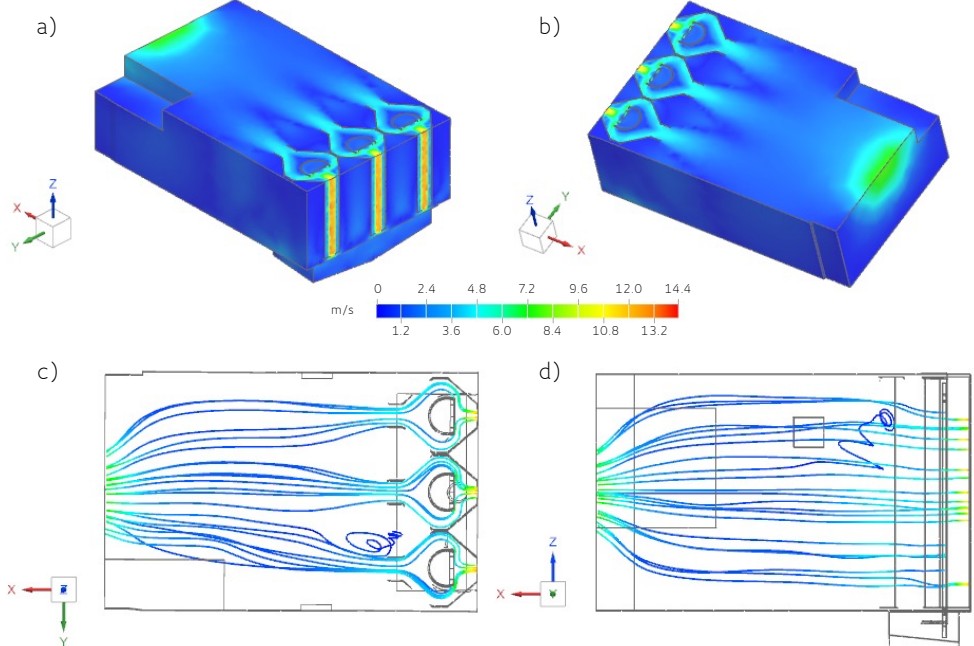

297

298 Figure 2. a) and b) Cutting plane in the flow velocity contour (in magnitude) in the case of an 8 m s$^{-1}$ air outlet flow
299 velocity; c) and d) set of streamlines in the collector (c- right view, d - top view) in the case of an 8 m s$^{-1}$ air outlet flow
300 velocity. Colour code indicates the different air velocity inside the collector.

301 Streamlines were also displayed (**Figures 2c and d** and **S7c and d**), with a set of seed points selected randomly

302 on the air inlet faces. They displayed velocity results by showing the path taken by a massless particle. Each point

303 along a streamline is always tangential to the velocity vector of the fluid flow. Again, the streamlines were only

304 slightly modified between the two velocities.

305 *Particle impact tracking*

306 Various classes of particles were injected into the collector at different air outlet velocities. **Table S2** lists the

307 number of water droplets for each air outlet velocity and each class of particles recorded by the solver in front of

308 the three inlets, represented by the three slots. Arbitrarily, approximately 60 000 particles are injected. We

309 calculated the number of injected droplets that impacted the vertical plates among the 60 000 particles; this allowed

310 the estimation of each class of particle and each velocity of the normalised efficiency of particle collection, as

311 reported in **Figure 3** in terms of the number of droplets and the mass of the droplets.



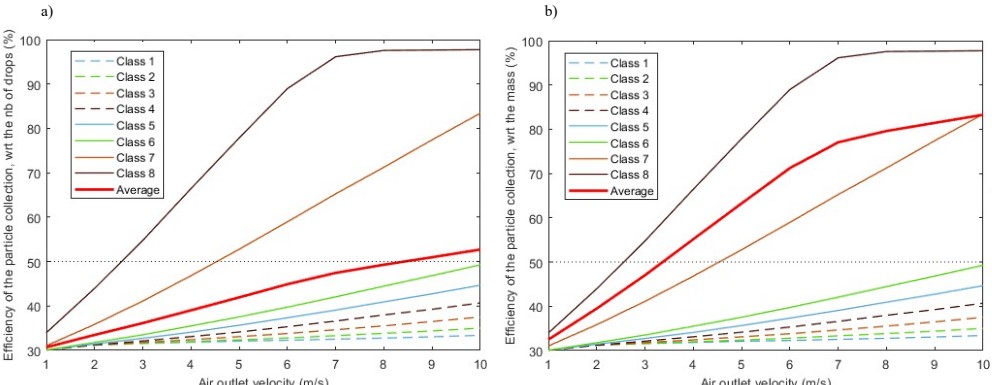

**Figure 3. Normalized efficiency of the particle collection, regarding the number of droplets (a) and regarding the mass of the droplets (b).**

We can observe that as the air outlet velocity increases, so does the collection efficiency for all droplet classes. For classes 7 and 8 (more than 15 µm in term of diameter), the collection efficiencies were >50% at low air outlet velocities (<5 m s⁻¹). At higher speeds, collection efficiencies >80% were achieved for both size classes. At the maximum speed, a collection efficiency of approximately 50% was reached for class 6 (10 µm in diameter). On average, for all droplet sizes, the average collection efficiencies of >50% in terms of numbers were achieved at air velocities >8 m s⁻¹. Considering the mass of the droplets, the two largest classes (7 and 8) naturally represented the largest mass of water collected. Because these two classes were efficiently collected even at low air velocities, a collection efficiency of 50% in terms of mass was achieved at 3 m s⁻¹ of velocity. At the maximum velocity, the average collection efficiency was approximately 80% in terms of mass.

The results highlight that the collector should be used with the highest velocity because the collection efficiency is theoretically maximal. However, at 7 m s⁻¹, we observed a slowdown in the overall collection efficiency because the largest drops were already 100% collected. These results allowed us to estimate the theoretical cutoff diameter, which represented the diameter at which 50% of the drops were collected. In our case, for the simulation conditions, this can be estimated at approximately 10 µm when the air outlet velocity is maximal.

These results are subject to limitations and uncertainties related to the modelled physical phenomena. First, the statistical results from the CFD simulations were based on a certain number of particles injected into the computational domain to achieve reasonable computing times. Second, the collection surfaces are supposed to be "ideal": a droplet, that impacts a plate, sticks to it; therefore, its transport by gravity to the funnel remains hypothetical. Finally, none of the physical phenomena were considered; the simulations were based on the equations of classical fluid mechanics, but other phenomena, such as electrostatics or Brownian motion, may affect the lightest particles. However, the performed simulations indicate good theoretical efficiency of the new BOOGIE collector for collecting cloud droplets, which also confirms that the distance between the air inlet slots and the outlet fan is adequate because it is beneficial for air flow stabilisation.

**3.2.2 Field sampling experiments**





To evaluate the performance of the BOOGIE sampler, 17 cloud events were collected at PUY station from May
to July 2016 and from July to November 2021. Seventeen cloud events, corresponding to twenty samples, were
collected using BOOGIE to evaluate its performance by measuring the collected water mass as a function of the
sampled volume of air (Wieprecht et al., 2005; Demoz et al., 1996). **Table S3** reports various parameters measured
during the sampling duration: meteorological parameters (temperature and wind speed) and microphysical cloud
properties (Liquid Water Content $LWC_{meas}$, and effective radius, $R_{eff}$, quantified by a PVM-100 probe et recorded
every 5 min).
First, we can estimate the cloud water collection rates of BOOGIE equal to $106 \pm 52$ mL h$^{-1}$. Water volume is
crucial because it determines the biological and chemical analyses that can be performed in the laboratory. The
BOOGIE collection rate allows sufficient cloud water to be obtained in a short duration, which is crucial because
the origin of the air mass that reaches the collection site can vary in a short time.
Experimentally, we can also evaluate the Collected LWC ($CLWC_{exp}$) in g m$^{-3}$ (Waldman et al., 1985) as:
$$CLWC_{exp} = \frac{M}{F \times \Delta t}$$ (1)
where M is the collected water mass (g); F is the sampler airflow (m$^3$ min$^{-1}$); and $\Delta t$ is the sampling duration (min).
To evaluate $CLWC_{exp}$, we estimated the sampler airflow of the new cloud collector. The optimal simulated
collection efficiency for this collector was simulated for an outlet air velocity equal to 8 m s$^{-1}$ corresponding to a
theoretical inlet air velocity of 12.8 m$^{-1}$ (Section 3.2.1). For cloud-water sampling, the sampler was operated at its
maximum inlet air velocity. Using a thermal anemometer directly in front of the slots, we measured an air velocity
of 14 m s$^{-1}$ (Section 3.1); therefore, we can estimate the outlet velocity at 8.75 m s$^{-1}$. To calculate the volume of
the sampled air, we use this value for the outlet air velocity; thus, the sampled air flow can be evaluated as follows:
with three inlets of 302 mm length and 12 mm width giving a total inlet surface of $10.9 \times 10^{-3}$ m$^2$ and an air velocity
of 8.75 m s$^{-1}$, then the airflow is 343.3 m$^3$ h$^{-1}$ (5.72 m$^3$ min$^{-1}$).
$CLWC_{exp}$ can be compared with the measured mean $LWC_{meas}$ for the 17 cloud events, as shown in **Figure 4**.



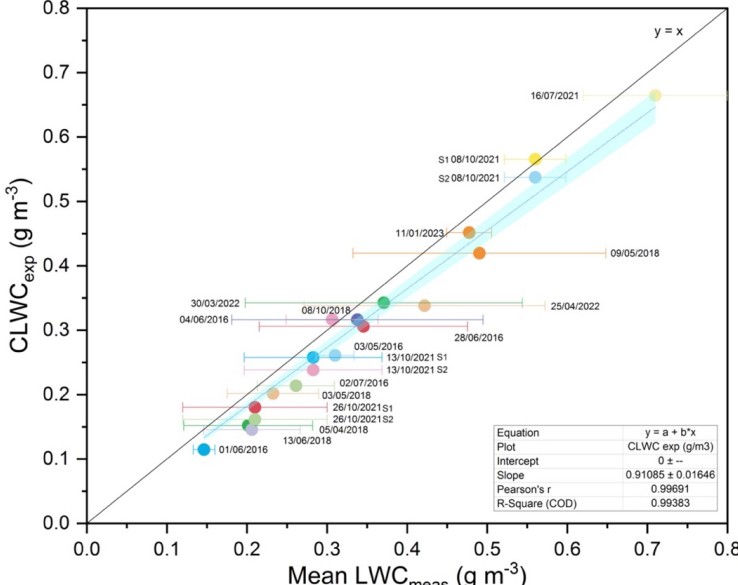


**Figure 4. Collected cloud water content CLWC$_{exp}$ vs measured LWC$_{meas}$ (in g/m$^3$) for a selection of 17 cloud events samples at the PUY station. Sampling dates and the standard deviation of the measured LWC are indicated. The black solid line represents the y = x function; linear fit of the experimental data with an intercept at 0 is represented by the dotted blue line and the blue area denotes the 95% confidence interval of this fit. S1/S2 corresponds to the same cloud event sampled with two BOOGIE collectors installed in parallel.**

The CLWC$_{exp}$ and measured LWC$_{mes}$ were well correlated (r$^2$ = 0.99; the slope of the linear regression was 0.91, and the intercept was 0 g m$^{-3}$). Systematic and random deviations from the "theoretical" efficiency are represented by a 1:1 line. Among the 17 cloud samples, only 2 cloud events presented a CLWC$_{exp}$ slightly higher than the LWC$_{meas}$.

The sampling efficiency can be estimated as follows:

$$\text{Sampling efficiency (\%)} = {}^{\text{CLWC}_{exp}}\!/_{\text{LWC}_{mes}} \times 100 \qquad (2)$$

The average calculated sampling efficiency over 17 cloud events was equal to 87.2 ± 8.6%. The sampling efficiency appeared to decrease when there was a shift to higher LWC$_{meas}$, which has also been observed with other samplers such as the CASCC2, possibly explained by interior collector wall losses for large droplets (Wieprecht et al., 2005). A plausible explanation for our sampler is the coalescence of very large droplets on the front face before aspiration into the slits.

The mean cloud wind speed and effective cloud droplet radius varied between the cloud events. **Figure S8** shows the sampling efficiency vs the three meteorological and microphysical parameters. The 17 clouds were sampled under conditions typically encountered at PUY for cloud sampling under warm conditions and for different seasons: temperatures >0 °C with a maximum value of approximately 11 °C; wind speed varying from 0.2 to 13 m s$^{-1}$. No tendency was observed between the sampling efficiency and temperature, supporting the fact that the collector can be operated over different seasons. The collector's orientation towards the wind is important, particularly under strong wind conditions. Incorrect orientation (*i.e.*, not in front of the wind) could drastically



reduce collection efficiency, whereas orientation towards strong winds could improve collection efficiency. For
the collected cloud events, we observed that the collection efficiency slightly decreased with wind speed; however,
the strength of the association was small (Pearson correlation coefficient of -0.23). At high wind speeds (gusts)
near 10 m s⁻¹, cloud droplet sampling can be non-isokinetic, explaining the possible perturbation of collection
efficiency. The average effective radius varied from 4.6 to 11 µm; there was no correlation between this parameter
and the collection efficiency, indicating adequate collection performance of the collector even for smaller droplets.
The collection efficiency calculated herein uses the theoretical total cloud water based on integrated measurement
methods (LWC). These estimates must be treated with caution because they are marred by several
errors/approximations listed here. These can be the result of the limitations of the instruments themselves (the
collector and the PVM probe) and the sampling conditions (wind); with the PWM-100 probe, we cannot optimally
capture the time evolution of the LWC because data are recorded every 5 min. Finally, the theoretical sampler
airflow used to calculate $CLWC_{exp}$ is intrinsically an estimate, and this can be additionally perturbed by the wind
condition. Nevertheless, this first comparison provides a rough estimate of the collection performance of the
BOOGIE collector, which appears to be suitable for contrasting environmental conditions.
**3.3 Comparison of cloud samplers**
A field campaign was conducted at PUY in 2016 to compare the new collector with other commonly used samplers.
The BOOGIE collector has been deployed to sample clouds together with the CWS used at the PUY station since
2001 and the CASCC2 (**Figure S9**). From 1ˢᵗ June to 2ⁿᵈ July, four cloud events were simultaneously collected
using these three samplers. The meteorological conditions and microphysical cloud properties were monitored
during the cloud events (**Figure S10**). Back trajectories were computed using the CAT model for the four cloud
events (**Figure S11**). The three samplers were oriented in front of the wind at the beginning of the sampling period;
changes in the wind direction were checked during this period, and the orientation of the collectors was modified
accordingly.
The prevailing winds during the first two cloud events (01 and 04/06/2016) arrived from the north-northwest and
north-northeast directions, whereas the other two (28/06/2016 and 02/07/2016) were locally associated with winds
coming from the southwest direction. This last event was also characterised by strong wind speeds of up to 14 m
s⁻¹ at the end of the sampling time. For the four cloud events, the wind directions did not drastically change during
the sampling duration except for on the 4ᵗʰ June where some fluctuations were observed; however, these were not
significant because the wind speed was extremely low (0.2 m s⁻¹). Regarding the microphysical properties, the first
cloud event presented lower mean measured LWC (0.15 g m⁻³) in comparison to the others (approximately 0.3 g
m⁻³). In contrast, the average radius was highest for the first cloud event (approximately 22 vs 9–13 µm in
diameter). The temperature corresponded to warm cloud conditions (between 6 and 10 °C), allowing the collection
of liquid droplets.
*Sampling efficiency*
First, the cloud water samplers were compared in terms of sampling efficiency, considering the calculated
$CLWC_{exp}$ and measured $LWC_{mes}$ (equation (2)). For the CASCC2, the sampled airflow was evaluated following
Demoz et al. (1996) (Section 2.3.2). The sampled airflow was evaluated for the CWS, which is a homemade
collector that follows the recommendations of Kruisz et al. (1993). As indicated in Section 2.3.2, the air inlet flow





velocity was measured as 13.5 m s$^{-1}$. Because the CWS and BOOGIE collectors have the same geometry as the
impaction system, we applied the ratio (1.6) evaluated for the BOOGIE collector to calculate the outlet velocity
(8.43 m s$^{-1}$). Therefore, considering the surface of the entry slot, the sampled air entering the CWS collector was
estimated to be equal to 113.9 m$^3$ h$^{-1}$ (1.90 m$^3$ min$^{-1}$).
**Table 1. Information on cloud water collection performed with BOOGIE, CWS and CASCC2 samplers for four**
**independent cloud events at PUY. The temperature, wind speed and $R_{eff}$ are averaged over the sampling time.**

| Cloud events: duration, mean temperature, mean wind speed & mean effective radius | Sampler | BOOGIE | CWS | CASCC2 |
|---|---|---|---|---|
| | Sampled airflow (m$^3$ h$^{-1}$/ m$^3$ min$^{-1}$) | 343.3/5.72 | 113.9/1.90 | 284/4.73 |
| Date = 01/06/2016 | LWC$_{mes}$ (g m$^{-3}$) | | 0.15 $\pm$ 0.01 | |
| Duration = 90 min | Sampled volume of air | 514.8 | 170.8 | 425.7 |
| T = 6.3 $\pm$ 0.2 °C | Collected water (g) | 59 | 19 | 40 |
| Wind speed = 8.1 $\pm$ 0.5 m s$^{-1}$ | CLWC$_{exp}$ (g m$^{-3}$)[*] | 0.115 | 0.12 | 0.094 |
| $R_{eff}$ = 10.8 $\pm$ 0.7 µm | Sampling efficiency (%)[*] | 79 | 76 | 64 |
| Date = 04/06/2016 | LWC$_{mes}$ (g m$^{-3}$) | | 0.31 $\pm$ 0.06 | |
| Duration = 180 min | Sampled volume of air | 1029.6 | 341.6 | 851.4 |
| T = 7.8 $\pm$ 0.2 °C | Collected water (g) | 326 | 110 | 261 |
| Wind speed = 0.3 $\pm$ 0.1 m s$^{-1}$ | CLWC$_{exp}$ (g m$^{-3}$)[*] | 0.317 | 0.322 | 0.307 |
| $R_{eff}$ = 6.6 $\pm$ 0.6 µm | Sampling efficiency (%)[*] | 103 | 105 | 101 |
| Date = 28/06/2016 | LWC$_{mes}$ (g m$^{-3}$) | | 0.35 $\pm$ 0.13 | |
| Duration = 60 min | Sampled volume of air | 343.2 | 113.9 | 283.8 |
| T = 9.3 $\pm$ 0.14 °C | Collected water (g) | 105 | 34 | 88 |
| Wind speed = 2.3 $\pm$ 0.4 m s$^{-1}$ | CLWC$_{exp}$ (g m$^{-3}$)[*] | 0.306 | 0.30 | 0.310 |
| $R_{eff}$ = 4.6 $\pm$ 1.0 µm | Sampling efficiency (%)[*] | 89 | 87 | 89 |
| Date = 02/07/2016 | LWC$_{mes}$ (g m$^{-3}$) | | 0.26 $\pm$ 0.05 | |
| Duration = 360 min | Sampled volume of air | 2059.2 | 683.3 | 1702.8 |
| T = 9.7 $\pm$ 1 °C | Collected water (g) | 440 | 135 | 290 |
| Wind speed = 12.0 $\pm$ 1.5 m s$^{-1}$ | CLWC$_{exp}$ (g m$^{-3}$)[*] | 0.213 | 0.198 | 0.170 |
| Reff = 6.1 $\pm$ 0.7 µm | Sampling efficiency (%)[*] | 82 | 76 | 65 |

[*] The collected LWC (CLWC$_{exp}$) is calculated following equation (1) and the sampling efficiency by equation (2).

The CASCC2 and BOOGIE samplers collected approximately 300 m$^3$ of air per hour, whereas the sampled volume
of air collected by the CWS was markedly lower (approximately 100 m$^3$ h$^{-1}$), which explains the lower amount of
collected water. The BOOGIE sampler presented a mean water collection rate for the four cloud events of 82 $\pm$ 32
mL h$^{-1}$ which was significantly higher than the two that of the other collectors (CASCC2: 62 $\pm$ 30 mL h$^{-1}$; CWS :
26 $\pm$ 11 mL h$^{-1}$) (t-test, p<0.05). On average, the calculated sampling efficiencies were 88 $\pm$ 11%, 86 $\pm$ 14%, and
79 $\pm$ 18% for BOOGIE, CWS, and CASCC2, respectively. Overall, the three collectors exhibited similar and
satisfactory collection efficiencies. This confirms that the volume of water collected by cloud samplers can be used
as a proxy to estimate cloud LWC. The slightly lower collection efficiency of CASSC2 may reflect the loss of
droplets off the strands and/or losses inside the collector on the walls, as highlighted by Wieprecht et al. (2005),
particularly for large droplets. This collector appeared to be more affected by the intensity of wind speed, with the
lowest collection efficiencies observed for the two windier cloud events. As reported by Kruisz et al. (1992) for





CWS and shown in this study for BOOGIE, no correlation of wind speeds to the $CLWC_{exp}$ of the samplers was
found. In the case of the 4th June cloud, the appearance of fine rain during sampling could possibly explain the
overestimation of collection efficiency observed for all collectors, as we did not observe conditions such as strong
winds that could disrupt the sampling.
Concerning the CASCC2, a sampling efficiency was previously determined during the FEBUKO experiments in
the Thüringer Wald (Germany) at 56 ± 17% (Wieprecht et al., 2005). Kruiz et al. (1993) calculated a sampling
efficiency of approximately 60% for the CWS during sampling experiments performed at Mount Sonnblick
(Austria). This sampling efficiency seems to be lower than that calculated in the present study. This could be
influenced by environmental conditions and cloud microphysical properties, which differ between collection sites.
The four cloud events have also been sampled at PUY under "optimal" conditions (summertime conditions with
limited wind speed and sufficient cloud LWC), possibly explaining the efficient collection of the samplers.
***Cloud water chemical and biological composition***
To compare the three cloud water collectors, we also focused on the chemical compositions of the three cloud
water samples collected in 2016. The concentrations of inorganic ions in samples collected with the CWS and
CASCC2 collectors (**Table S4, Figure S12**) were compared to the concentrations measured in samples collected
with BOOGIE using the discrepancy factor ($D_f$) calculated using **equations 3a and 3b**.
$$D_{f,CWS} = \frac{C_{BOOGIE}-C_{CWS}}{\left(\frac{C_{BOOGIE}+C_{CWS}}{2}\right)} \tag{3a}$$
$$D_{f,CASCC2} = \frac{C_{BOOGIE}-C_{CASCC2}}{\left(\frac{C_{BOOGIE}+C_{CASCC2}}{2}\right)} \tag{3b}$$
where $C_{BOOGIE}$ is the concentration of ions measured in samples collected with BOOGIE, and $C_{CWS}$ and $C_{CASCC2}$
are the concentrations of ions measured with CWS and CASCC2, respectively.
**Figure 5** shows the estimated $D_{f,CWS}$ and $D_{f,CASCC2}$ for anions and cations for cloud samples. The horizontal dashed
lines represent the analytical error on the measurement, which is comparable with $D_{f,CWS}$ 02/07/2016 for sulphate,
nitrate, chloride, and ammonium and $D_{f,CASCC2}$ 28/06/2016 and 02/07/2016 for nitrate, sulphate, chloride, and
sodium. The other $D_f$ values were higher, but generally <0.5, which could represent a good comparability of the
cloud collectors, because the chemical composition of cloud condensation nuclei may be inhomogeneous. At first
glance, concentrations with the CASCC2 appear to be slightly higher, but not for all ionic species and not for all
the cloud events. These three samplers present specific designs and surfaces of collection (plate for BOOGIE and
CWS vs strands for CASCC2), leading to different estimated cutoff diameters (10 μm for BOOGIE, 7.5 μm for
CWS, and 3.5 μm for CASCC2) and possibly to small differences in the chemical composition of the samples.



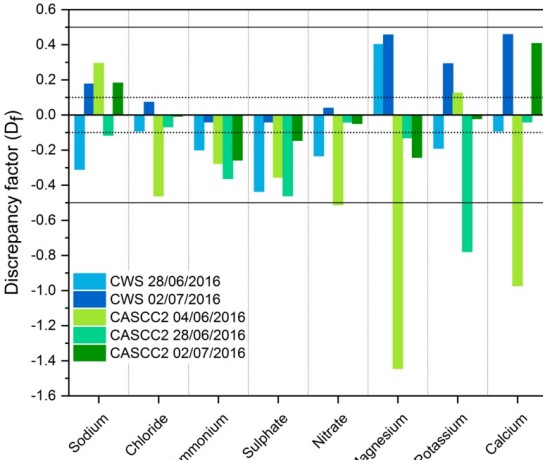


**Figure 5. Histograms presenting discrepancy factors ($D_f$) between BOOGIE and CWS and CASCC2 calculated using anion and cation concentrations for the three cloud samples. The dashed lines represent the analytical error, whereas the plain line represents the 50% discrepancy.**

Formaldehyde and hydrogen peroxide concentrations have been also measured in samples obtained with the three
collectors. Concentrations and discrepancy factors between collectors are presented in **Figure 6**. These results are
consistent with what was observed with the ionic content because the collectors indicate $D_f$ values mostly within
the analytical error and maximum measured $D_f$ values <0.5.

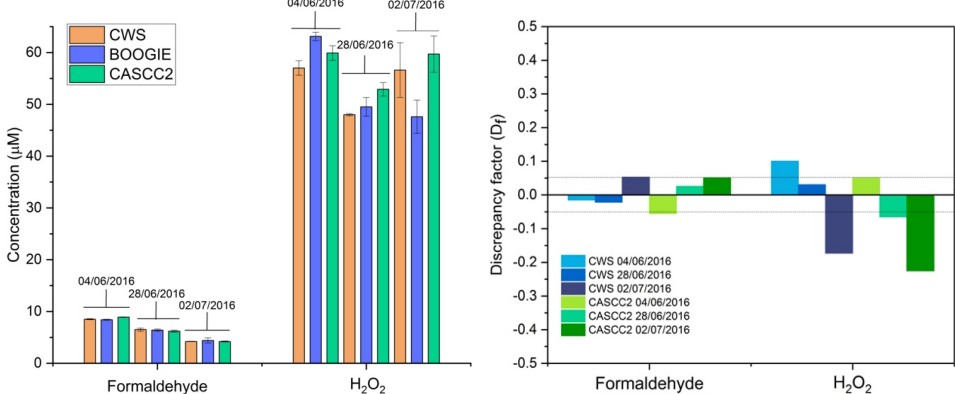

**Figure 6. Left: Histograms presenting the anion and cation concentrations for the three cloud samples collected using**
**CWS, BOOGIE, and CASCC2 in parallel. The error bars correspond to the standard deviation. Right: Histograms**
**presenting discrepancy factors ($D_f$) between the BOOGIE and CWS and CASCC2. The dashed lines represent the**
**analytical error.**
To further evaluate BOOGIE, two identical collectors were installed at the PUY station in 2021 to check for
differences in the chemical composition of cloud waters collected in parallel. For clouds on 08/07/2021, chemical
measurements were performed in triplicate to analyse the statistical differences (**Figure S13, Table S5**). The error
bars depict the analysis error, which is higher than the discrepancy between the BOOGIE collectors for sodium,
potassium, calcium, and chloride. The black plain line represents the p-value obtained for the t-test (right y-axis);
if the p-value is <0.05, represented in the plot by the yellow dashed line, the difference between the two BOOGIE



collectors is significant, as observed for magnesium, nitrate, and chloride. Nevertheless, the difference was not
significant for sodium, ammonium, potassium, calcium, and sulphate, indicating good reproducibility of sampling
with the BOOGIE collectors.
Given the uncertainties in laboratory measurements and the possible intrinsic variability of the chemical
composition within the cloud system, we can reasonably argue that the chemical compositions of the collectors
are comparable. Schell et al. (1992) compared two single-stage cloud impactors with different designs and
highlighted the large differences between the ionic compositions of the samples. These differences have been
discussed to be related to different microphysical properties of the sampled clouds that induced bias in the
collection: smaller droplets can be sampled with a lower cutoff diameter of the collector, and a lower LWC can
eventually induce some evaporation of the smaller droplets. The three cloud events presented "stable"
microphysical properties during their collection period (**Figure S9**). This could explain the good agreement
between the collectors in terms of their chemical composition. Wieprecht et al. (2005) compared the chemical
composition of cloud water collected with a low-volume single-stage slit jet impactor and with the CASCC2 string
collector and reported 8–15% differences in the solute ionic mass in cloud water, in the range observed in the
present study (4–35% of differences, average of 12%) between the three collectors.
The microbial energetic state given by the in-cell ATP and ADP concentrations from each cloud sample was
assessed during the inter-comparison campaign (see Supplementary Information for a description of the protocol).
The ATP/ADP ratio gives the energetic stress of the cloud water microbiota; a ratio <0.6 indicates a good energetic
state, 0.6 to 1, a medium one, and >1, a low energetic state. The measured ratios are listed in **Table S6**. The
ATP/ADP ratio ranged from 0.2 to 0.4, revealing a good energetic state of microflora for each sample. The
measured ATP/ADP ratios were similar for the cloud water samples from the three collectors. Thus, we argue that
the three samplers could be considered non-stressful and suitable for cloud microbiota collection.
**4 Conclusions**
This study presented a new cloud collector called BOOGIE. This single-stage collector allows cloudy air
containing aqueous droplets to be drawn through three air inlets in the form of vertically oriented slots. The cloud
droplets were collected using vertical plates placed behind the slots, allowing them to be impacted. They then
flowed by gravity along the plates, fell into a funnel, and ended up in a sterilised glass bottle. It was made of
aluminium, but can be manufactured from other materials, such as plastic materials such as nylon or PTFE to
investigate transition metal ions in cloud waters. The cloud collector can be connected to the mains or run on
batteries (12 V voltage); thus, the collector can be operated at its own power during field measurement campaigns
for at least 4 h using a 2 kg small battery. Parts of the sampler were removed for cleaning; the front face, impaction
chamber, funnel, and glass bottle were sterilised in an autoclave. This allowed for the characterisation of the
biological content of the sampled clouds (biodiversity, concentration, and viability/activity) (Vaïtilingom et al.,
2012). Biological and chemical collector blanks were easily prepared by spraying MilliQ water onto the collection
plates and collecting the water flowing into the collection glass bottle.
CFD simulations were performed to investigate how the collector captured cloud droplets. First, considering the
3D-dimensional structure of the collector, some turbulences were simulated inside the collector, which was
reassuring. Different classes of cloud droplets were injected into the collector to simulate their impacts on the





collection plates. This theoretical study indicates that on average, for all droplet sizes (radius from 2.5 to 10 μm),
the average collection efficiencies of >50% in terms of numbers were achieved at air outlet velocities >8 m s$^{-1}$. A
collection efficiency of approximately 50% was reached for 5 μm droplets in a radius that gave us an estimate of
the 50% cutoff diameter of the collector (approximately 10 μm). This estimate seems slightly higher than the cutoff
diameters of other cloud samplers (more in the range between 3.5 and 10 μm in diameter). However, comparisons
of cutoff diameters between samplers are difficult because these estimates are made using different methods; in
particular, the theoretical collection efficiency often considers the Stokes number (Demoz et al., 1996).
Based on the 17 cloud events sampled at the PUY station, a mean water collection efficiency was calculated as
156 ± 52 mL h$^{-1}$ for clouds presenting various microphysical cloud properties: the mean LWC was between 0.15
and 0.71 g m$^{-3}$ and the mean effective radius R$_{eff}$ was between 4.6 and 11 μm. This made it possible to obtain
sufficient water volumes over short periods for targeted chemical and biological analyses. This is crucial for
minimally integrating the cloud properties in space and time. Methodological developments in recent years have
made it possible to assess the organic composition and biodiversity of this aqueous environment using non-targeted
methods (Rossi et al., 2023; Bianco et al., 2018). This requires large volumes of cloud water (hundreds of milliliters
or even liters of water), which can be collected rapidly using the new collector alone or by duplicating it.
Considering the measured LWC, $LWC_{meas}$, the sampling efficiency of this new collector was estimated at 87.2 ±
8.6% over the same set of cloud events collected at PUY. No significant decrease in the collection efficiency was
observed as the wind speed increased, over the range of variation between 0.3 and 13 m s$^{-1}$. No significant
correlation was observed between the efficiency and mean measured effective radius. A low LWC cloud event
would present a greater proportion of liquid water residing in smaller droplets; therefore, for a low LWC, we
expected the collection efficiency to diminish owing to the cutoff diameter. However, this decrease was not
observed in the cloud samples. Additional measurements of droplet size distribution during sampling would be
beneficial for clarifying this issue.
This new cloud water collector was compared with two other single-stage collectors that are commonly used by
the scientific community to study cloud composition and environmental variability. We selected the CWS initially
developed at the University of Vienna (Kruiz et al., 1993) and often deployed at mountainous sites such as Mount
Sonnblick (Austria) and the PUY station. The impaction of the droplets occurs in a metallic plate horizontally
installed in the collector, and it can be sampled under supercooled conditions. The other collector was one of the
samplers developed by the California Institute of Technology (Caltech) for studies on fog and clouds (Daube et
al., 1987), the CASCC2. This active sampler is a compact version of the CASCC, in which droplets are collected
by impaction on a set of six rows of stainless-steel strings; it is highly efficient in terms of collection and is not
affected by raindrops owing to its design. It cannot function under supercooled conditions. The proposed BOOGIE
collector aims to efficiently sample cloud droplets under warm cloud conditions and is designed to be easily
sterilisable. Under low wind conditions, it is not affected by rain.
We compared the collection efficiency and chemical compositions of these three collectors. For the four studied
cloud events, the BOOGIE collector presented an elevated water collection rate of 82 ± 32 mL h$^{-1}$ (CASCC2: 62
± 30 ml h$^{-1}$; CWS: 26 ± 11 mL h$^{-1}$). This can be explained by the increased volume of cloudy air entering the new
collector. On average, the calculated sampling efficiency was 88 ± 11% for BOOGIE, in the same range as that
for CWS and CASCC2. The chemical and biological compositions measured in the samples collected by the three



collectors can be evaluated as comparable; however, some differences can be highlighted, which can be explained
by the design of the collector, type of collection, and inhomogeneous chemical composition of the cloud
condensation nuclei.
This new BOOGIE collector is designed for use in field campaigns and long-term observatory sites. It contributes
to the evaluation of the complex cloud water bio-physico-chemical composition, to the analysis of its
environmental variability; it allows a sufficient volume of water to be collected to characterize the chemical and
biological transformations occurring in it. This will help better constrain detailed cloud chemistry models that need
to be validated (Barth et al., 2021). For future development, our team aims to reduce the size and weight of the
collector such that it can be installed under a native balloon. The second development concerns the automation of
this collector to initiate collection remotely and increase the sampling frequency. Finally, we aim to conduct
intensive campaigns in the frame of the ACTRIS "Cloud In Situ" network to compare the collectors used by the
scientific community at other measurement sites.
*Data availability:* All data are available through communication with the authors.
*Author contributions:* LD, MV were responsible of the project. MV, CBern and LD designed the new instrument,
MR created the 3D plans of BOOGIE. CBert performed the CFD analysis. MV, AB and LD conducted the cloud
sampling. MV and AB performed the chemical and biological analysis in the lab. LD and MV performed the data
analysis. LD, MV and AB conducted scientific analyses. LD prepared the manuscript and designed the figures,
with contributions from all authors. MV, AB, MR, CBert revised the manuscript.
*Competing interests.* The authors declare that they have no conflict of interest.
*Acknowledgments.* This study on cloud water characterisation was performed in the framework of the CO-PDD
instrumented site of the OPGC observatory and LAMP laboratory. This study was supported by the Université
Clermont Auvergne, Centre National de la Recherche Scientifique (CNRS), and Centre National d'Etudes
Spatiales (CNES). The authors are also grateful for the support from the Fédération des Recherches en
Environnement through the CPER funded by Region Auvergne–Rhône-Alpes, the French Ministry, ACTRIS
Research Infrastructure, and FEDER European regional funds. The authors also thank I-Site CAP 20-25. We thank
Olivier Masson from the IRSN for their CASCC2 collector, which was gratefully lent during the inter-comparison
campaign.
*Financial support.* The authors are grateful to the Agence Nationale de la Recherche (ANR) for its financial
support through the BIOCAP (ANR-13-BS06-0004) and METACLOUD (ANR-19-CE01-0004) projects. The first
project has financed the work of Mickaël Vaïtilingom during his post-doc at the LAMP laboratory and the second
one allowed for their evaluation for specific scientific questions. We thank OPGC for additional funding and
OPGC Service de developpement technologique for manufacturing the cloud samplers. The Institut de Chimie de



Clermont-Ferrand and Laboratoire Microorganismes: Génome Environnement laboratories are acknowledged for
allowing access to their chemical and microbial analytical platforms.

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
