# Peer review of "Design and evaluation of BOOGIE: a collector for the analysis of cloud composition and processes"

_Atmospheric Measurement Techniques, 2024_

## Author Comment (AC1)

**Reviewer 1:**

This manuscript is a very elaborate description of an active fogwater collector, which is employed at the Puy de Dôme site in France. The collector is precisely described, its internal flow is modelled with a CFD fluid dynamic model, the collector and its performance are tested in the field during side-by-side comparisons with two other fog collectors. Results are described in great detail, good performance and good agreement with the other 2 active collectors is shown. This refers to the sampling efficiency (with respect to the liquid water content (LWC) of the fog or cloud, and to the results of inorganic ion analyses. The manuscript is well written, well supported by data (including the supplementary material) and deserves publication in AMT after the following 3 concerns have been addressed accordingly.

We would like to thank the reviewer for his review and the high regard in which it holds the development work that was carried out. We answer to his/her comment below in blue.

1- Results of 17 cloud events are presented. Is that number (17) the entire population of events probed? If yes, please state that prominently in the manuscript. Or, alternatively, are these 17 "beautiful" events that were selected from a larger pool of data. If so, authors are asked to communicate in the manuscript the arguments / parameters / thresholds they used to select these 17 events. Important: At some point in the manuscript, sampling dates (and times) of all 17 events need to be documented.

Yes, you are right we must explain why 17 cloud events were selected for our analysis. The collector was developed in 2016 and between 2016 and 2023, more than 17 cloud events were sampled. We selected these events based on the availability of LWC measurements and of the exact measured mass of the collected water. Table below presents all the events between 2016 à 2023 that we excluded from our analysis based on these criteria. We added in the manuscript a sentence to explain how the 17 events analyzed in this work were selected (section 3.3.2). To answer to your other comment, the information regarding the sampling dates (and times) of the 17 cloud events are indicated in Table S3 in the SI together with meteorological parameters (T, wind speed), microphysical parameters (liquid water content [LWC], effective radius) and mass of the collected cloud water.

| Date of sampling | LWCmeas | Measured Mass of water (g) |
|---|---|---|
| 05/06/2016 | OK | N.A. |
| 12/07/2016 | OK | N.A. |
| 13/10/2016 | OK | N.A. |
| 14/10/2016 | OK | N.A. |
| 21/10/2016 | OK | N.A. |
| 26/10/2016 | OK | N.A. |
| 21/02/2017 | OK | N.A. |
| 08/03/2017 | OK | N.A. |
| 24/03/2017 | OK | N.A. |
| 02/05/2017 | N.A. | |
| 19/05/2017 | N.A. | |
| 29/06/2017 | N.A. | |
| 31/08/2017 | N.A. | |
| 26/09/2017 | N.A. | |
| 03/10/2017 | N.A. | |
| 24/08/2018 | N.A. | |
| 14/09/2018 | N.A. | |

| | | |
|---|---|---|
| 01/10/2018 | OK | N.A. |
| 08/10/2018 | OK | N.A. |
| 02/03/2019 | N.A. | |
| 15/03/2019 | N.A. | |
| 25/09/2019 | N.A. | |
| 02/10/2019 | N.A. | |
| 22/10/2019 | N.A. | |
| 11/03/2020 | N.A. | |
| 28/04/2020 | N.A. | |
| 17/07/2020 | OK | |
| 10/11/2020 | OK | |
| 19/11/2020 | OK | |
| 28/04/2021 | OK | N.A. |
| 06/05/2021 | OK | N.A. |
| 08/07/2021 | OK | N.A. |
| 16/09/2021 | OK | N.A. |
| 03/11/2021 | OK | N.A. |
| 17/02/2022 | N.A. | |
| 17/03/2022 | OK | N.A. |

N.A. : Not Available.

To address the reviewer's comment, we decided to add new cloud events to this study to improve our collector evaluation study. So, 4 cloud events collected in spring 2024 corresponding to 19 samples were added to the database analyzed in this article. Some of the events were sampled in windy conditions, enabling us to further analyze the influence of wind on the BOOGIE collector efficiency.

2- The title of the manuscript is awkward and needs revision. First, BOOGIE seems to be a nickname for the collector, it does not appear to be an acronym. However, the prominent mention of the word (capital letters in the title) suggests something extraordinary which is not present.

To be honest, the collector's name "boogie" is a tribute to an "entity" that has been important in cloud-related activities on the Puy de Dôme site. Of course, linking this acronym to the description of what the collector does is rather difficult. But our aim was not to oversell what our collector can do, since it only collects water! But collecting cloud waters among various environmental conditions is crucial and subsequent analysis in the lab helps to better assess the effect of clouds on atmospheric chemistry.

Also, authors mention that the collector is newly designed (for example, in line 571), which is not really true since it has been employed for over 30 years.

Before 2016, we used a CWS sampler following the design proposed by Kruiz et al. (1993). The collector (BOOGIE) that is presented in this work has been developed in our lab in 2016. We only used the BOOGIE collector during the last 8 years at the Puy de Dôme station and also at La Réunion Island. This is indicated in the introduction section (lines 163-164). We modified the text to avoid any confusion about the novelty of this development.

Further, the title of the manuscript promises more than the manuscript really offers. Of the second part of the title (…Biological, Organics, Oxidants, soluble Gases, inorganic Ions and metal Elements), this reviewer really only found inorganic ions. Biological analyses are mentioned, but results are not shown

in a very limited way: ATP/ADP ratio data are shown in the supplementary, results briefly discussed in the main manuscript (lines 506 – 512). Oxidants and metal elements are not even mentioned in the results section. The latter would require special attention before the backdrop of the metal construction of the collector. Very few results are shown for formaldehyde and hydrogen peroxide, while it is not clear to this reviewer if they meant to represent organics or soluble gases. All in all, the second part of the title needs to be removed.

Yes, we agree with your comments. The chemistry results focus mainly on inorganic ion chemistry, as well as measurements of: ATP/ADP, 2 oxidants ($H_2O_2$, nitrate) and one specific organic compound (formaldehyde).

We agree that the section about the ATP/ADP is brief, it is because the energetic cell states were comparable and easy to interpret. The ATP/ADP ratio were similar between the three samplers for each cloud event. The cloud microflora has an excellent energetic state for each collector sample; this indicates that these three samplers were not stressful for the microbiota and keep the microbial cellular integrity during the sampling.

The idea of the paper was not to make an in-depth comparison of the measurements obtained between different collectors but to mention that the measurements obtained were consistent with each other. The main objective of the paper is to present the collector and its efficiency. Moreover, within the framework of the European ACTRIS network, a special effort is being made on these aspects: intercomparison campaigns between collectors are being conducted (in which BOOGIE has been deployed) to compare their collection efficiency, and intercomparisons between cloud chemistry measurements are also being carried out. This work is still in progress, but initial results confirm that our collector is efficient, and indicate that laboratory measurements (mainly ions) are comparable between collectors, even if some bias may appear. This work will be promoted within ACTRIS (see The Centre for Cloud Water Chemistry (CCWaC): https://www.actris.eu/topical-centre/cis/centre-cloud-water-chemistry-ccwac).

Since we do not present the analysis of all the chemical compounds indicated in the title, we propose removing the second part of the title but keeping the proposed name BOOGIE. We hope this will be acceptable to the reviewer.

3- Droplet size distribution (DSD) data are not shown or analyzed. It is suggested (lines 550, 551) that such data was not available. On the other hand, it seems that there is plenty of literature from that site showing and discussing DSD data. Authors are asked to clearly state in the manuscript that such data is not available in all 17 events. Alternatively, please make very strong arguments in the manuscript about the reason why such data was not employed in this manuscript. Alternatively, analyze such data to support arguments of collection efficiencies as a function of droplet sizes.

To estimate the sampler efficiency, we used data from the PVM-100 instrument installed on site. It allows to evaluate the particle volume density (or LWC: liquid water content) and the particle surface area density (PSA). The effective radius $R_{eff}$ can be calculated using LWC and PSA. This instrument has been in use at Puy de Dôme since the very beginning of cloud studies and is listed as mandatory instrument in the frame of the topical center "cloud in situ" measurements of the ACTRIS program. Measurement of LWC by the PVM is sufficient to assess collection efficiency of a collector, and other studies in the past have also used the data produced by this instrument to study their samplers (Kruiz et a., 1993; Demoz et al., 1996).

Of course, an instrument such as a fog monitor (FM-120) or particle measurement system (FSSP-100) designed for use during ground-based studies and allowing real-time display of particle concentration could give additional information to better interpret the role of the calculated cutoff diameter using CFD

simulations on the collection efficiency. But unfortunately, measurement of droplet size distribution was not available for all the 17 studied cloud events presented.

As you mentioned, numerous instruments have been deployed at Puy de Dôme (Guyot et al., 2015) with the aim of intercomparing instruments for measuring cloud microphysical properties (PVM-100, FSSP, FM-100, cloud droplet probe (CDP)…). This study highlighted the need to use an instrument such as the PVM that provides bulk measurement of cloud microphysical properties; indeed, this allows to evaluate the size distributions measured by instruments measuring size distribution, as the latter are highly sensitive to wind and cloud orientation.

---

## Author Comment (AC2)

**Reviewer 2**

The authors present design information, along with select performance evaluation information, for a new cloud collector which they call BOOGIE. The collector is designed based on inertial plate impaction behind 3 rectangular jets and features an air sampling rate in the range of several existing collectors. For example, the air sample rate, as pointed out by the authors is slightly higher than the CASCC2 but remains well below the air sample rate of the original CASCC. The authors use CFD to estimate droplet collection performance and compare sample collection efficiency via comparison to optically measured cloud LWC and sample composition to measurements from two widely used samplers, the CWS and CASCC2. Overall, the manuscript is well organized and the new collector design and evaluation should prove of interest to AMT readers. There are, however, several items in need of attention before the manuscript should be considered for publication including one critical error that influences several analyses.

We would like to thank the reviewer for his review and the high regard in which it holds the development work that was carried out. We answer to his/her comment below in blue.

Major items:

1- I was surprised to see a jet impactor operating using a rotary bladed fan since jet impactors often have a relatively high pressure drop and the performance of these fans can strongly decrease at increasing pressure drop. It appears that this issue is mitigated by the relatively high drop size cut of the collector (10 um). The manuscript would be improved by a discussion about the pressure drop generated in the collectors and how this relates to the ability to collect small droplets. Why was a 10 um size cut targeted rather than a more conventional value around 5 um? Was this a design choice or just how things turned out?

We initially developed the collector based on the geometry from the CWS we used at the puy de Dôme station. Therefore, we decided to maintain the distance between the inlets and the impaction plate, and the size of the inlets where the air is drawn is also conserved. The idea was to benefit from a validated, easy-to-clean collection system, particularly for measuring the biological component. We also duplicated the number of air inlets (3) and placed the impaction plates in a vertical position. Secondly, the suction system effectively uses a rotary bladed fan (like the CASCC) to aspirate the cloud droplets into the collector. We carried out several experiments in the field to test its collection rate; these tests were conclusive.

Concerning the pressure drop inside the collector, of course it exists. We decided to use this rotary bladed fan as it allows the air to be drawn into the collection system at high-flow rate. This type of fan is also used by the CASCC2 sampler for the same reason we presume. Moreover, even if there is a pressure drop (flow rate at 600 m$^3$/h in theory vs. 433 m$^3$/h measured experimentally), the volume of air is still large, so collection rate of water is effective. The dilemma relies in:

- we draw in high volumes of air, but the cut-off diameter of the sampler may be higher;
- we use systems with smaller cut-off diameters, but we must increase the sampling time to get enough volume for analysis.

After testing the collector under real conditions, we then decided to carry out CFD simulation to study how the droplets were collected in the system. This theoretical calculation enabled us to analyze how the air circulates inside the system and to assess which particles were most efficiently collected. By this way, we demonstrated that few turbulences occur around the collection plates and by injecting droplets we add the possibility to estimate a theoretical cut-off diameter around 10 μm, which was reassuring.

There are different ways to calculate a cut-off diameter using for example fluid mechanics (i.e., Stockes number as described in Demoz et al., 1996 and used for example recently in Du et al., 2023). But, to our opinion, it is quite difficult to compare our estimates by simulations to these calculations. Moreover, in the CFD simulations, we only consider a constant flux of the air leaving the collector since it is impossible to reproduce the fan's rotational movement.

In the new version of the manuscript, we experimentally evaluated the exit velocity of the air flow even if it is difficult to measure as the reviewer mentioned below. We decided to do that since the measurement of the air inlet velocity presented also some limitations. By this we estimated the air flow rate that is needed to estimate the collection efficiency of the collector. To understand how the air flows inside the collector and given that estimating the air flow in the sampler suffers from certain limitations, we decided to carry out numerous CFD simulations, considering different air flow velocity from 2 m/s to 10 m/s at the outlet.

Regarding the pressure drop inside the collector, we add in the new version of the manuscript this paragraph:
"The pressure drop in the BOOGIE impactor can be estimated from the fan and flow characteristics. Since the flow rate has been calculated at 433 m³ h⁻¹, the pressure drop compensated by the fan is estimated at 220 Pa, and consequently the pressure drop in the impactor is around 210 Pa. The variation in density is less than 0.0025 kg m⁻³, i.e. a variation of less than 0.25%. The flow can be considered incompressible, and conservation of flow-volume can be used. The average velocity at the BOOGIE inlet is estimated at 11 m s⁻¹, by dividing the flow by the inlet cross-section of 10.9 10⁻³ m². This average velocity differs from the measured velocity at inlet (14 m.s⁻¹) due to the velocity profile at the slots. The measurement corresponds to a maximum velocity."

The pressure drop is therefore rather low. Compared to atmospheric pressure, the variation is less than 0.25%, which has no influence on drop collection (no change in velocity, additional evaporation, etc.). The flow rate of 600 m³/h is where there is no pressure drop. This is the maximum flow rate supplied by the manufacturer. To our opinion, there is no relationship between cut-off diameter and pressure drop. The pressure drop is linked to pressure losses, due to air friction. The collection effect is linked to the difference in inertia between the air and the water droplets. It's linked to the air speed and its changes in direction. The more abrupt the changes in air direction, the less the drops (especially the larger ones) follow the air's trajectory and can collide with surfaces. High velocity induces more pressure loss and therefore a greater drop in pressure.

References :
Demoz, B. B., Collett, J. L., and Daube, B. C.: On the Caltech active strand cloudwater collectors, Atmos. Res., 41, 47-62, https://doi.org/10.1016/0169-8095(95)00044-5, 1996.
Du, P., Nie, X., Liu, H., Hou, Z., Pan, X., Liu, H., Liu, X., Wang, X., Sun, X., Wang, Y.: Design and Evaluation of ACFC—An Automatic Cloud/Fog Collector, Atmosphere, 14, 563, 2023.

2- It is strange that the authors often present results in terms of their relationship to the exit velocity. Exit velocities behind a rotating fan are difficult to measure. Further, the relationship between flow entering the collector and flow exiting the collector depends on the pressure drop through the collector.
3- Lines 164-165: The authors' statement here about the theoretical flows entering and exiting the collector violates the principle of conservation of mass. The entering and exiting volumetric flow rates should be the same when adjusted for pressure drop through the collector. I think the theoretical ratio here is meant to refer instead to velocities.

Responses to comments 2 and 3:

Initially, we experimentally measured the air velocities of our collector at the system inlet, i.e. in front of the slots. We did this on the 3 air inlets and at different heights. We also did this by modulating the intensity of the fan. This is shown in figure S6 in the manuscript. By this way, we characterized the air flow inlet velocities for different fan intensities. This helped us to estimate the air flow entering the system that is a crucial parameter to investigate the collection efficiency of our new collector.

Next, we carried out numerical CFD simulations, keeping the geometry of the collector in its entirety. To reproduce how the air enters our system we need to constrain the air outlet flow on the rear face of the collector to reproduce the air inlet flow measured experimentally at the system inlet. We simulate air outlet flow at the fan considering the surface area of the fan; we do not reproduce the fan's rotation. We initially to not attempt to experimentally measure velocities behind the rotary fan. We did several CFD simulations to numerically estimate how each size class of droplets were collected. Regarding the areas of the slots and of the fan, based on the mass conversion, there was a theoretical ratio of 1.6 ratio between the inlet air volume flow and the air outlet volume flow. As highlighted by the CFD simulation, the air inlet velocity at the slot level is highly heterogenous. This led us to question the relevance and robustness of air velocity measurements at the slot level.

Therefore, we decided to also measure the outlet air flow velocity even if, as pointed by the reviewer, it is difficult to measure. By this way, our objective was to give a more robust estimate of the sampler airflow that is a crucial parameter to estimate the collected LWC ($CLWC_{exp}$) (eq. 1 in the manuscript). This will also help us to validate the simulated values performed by CFD regarding the inlet/outlet velocities.

We added in the manuscript the following paragraph in the new section "3.2 Evaluation of the air flow inside the BOOGIE collector" describing the experiment we conducted to estimate the sampler air flow: "[...] we designed an experiment to measure the air flow at the collector outlet. The airflow rate at the fan outlet was measured using the following procedure. A 3.5 m long PVC pipe with an internal diameter of 154 mm was installed after the fan outlet. This diameter enables the entire flow to be measured without reduction, thus limiting the additional pressure losses generated by the addition of the pipe. A hot-wire anemometer was installed in the tube at 3 m from the fan. The large distance/diameter ratio (greater than 19) minimizes disturbances (high turbulence and vortex rates) as the air passes through the axial fan.

The flow velocity profile is measured every 5 mm along the diameter. Flow rate is calculated by summing the average velocity for each ring by the ring area. The flow rate was estimated at 433 m³ h⁻¹ at 90% of the fan speed. The average velocity in the pipe is found by dividing the flow rate by the cross-sectional area, which corresponds to a velocity of 6.5 m s⁻¹. Based on this velocity, the Darcy-Weisbach formula and the Moody diagram (with a relative roughness of 2 10⁻⁵), the pressure drop in the pipe is estimated at 10 Pa. As a result, the addition of the pipe has little influence on the flow rate.

The pressure drop in the BOOGIE impactor can be estimated from the fan and flow characteristics. Since the flow rate has been calculated at 433 m³ h⁻¹, the pressure drop compensated by the fan is estimated at 220 Pa, and consequently the pressure drop in the impactor is around 210 Pa. The variation in density is less than 0.0025 kg m⁻³, i.e. a variation of less than 0.25%. The flow can be considered incompressible, and conservation of flow-volume can be used. The average velocity at the BOOGIE inlet is estimated at 11 m s⁻¹, by dividing the flow by the inlet cross-section of 10.9 10⁻³ m². This average velocity differs from the measured velocity at inlet (14 m s⁻¹) due to the velocity profile at the slots. The measurement corresponds to a maximum velocity.".

In the previous version of the manuscript, based only on the air inlet measurements, we did calculations that were false as pointed by the reviewer due to the non-respect of the conservation of mass. Note that, in the CFD calculation, of course we do not violate the principle of conservation of mass.

We therefore re-calculate all the collected LWC ($CLWC_{exp}$) and we redid the figure 4 that present experimental collected LWC vs the measured LWC. The collection efficiency (in %) were also recalculated and are indicated in Table S3 in the Supplementary Information. We also decided to include in this study additional clouds collected at the puy de Dôme, notably in spring 2024 during an international RACLET measurement field campaign as part of the ATMO-ACCESS program. 4 cloud events corresponding to 19 samples were added to our analysis.

4- I am somewhat surprised that some fraction of droplets impacted on the vertical plates are not blown off the sides of the impaction plates, pulled by the airflow around those plates. The collection efficiency results suggest this is not, however, a major issue. Can the authors comment on what prevents this from happening? Does the CFD simulation suggest there is a quiescent stagnation zone near the plate surface that prevents the drops from being pulled toward the plate edges?

For the calculation of droplet collection, in the CFD simulation, when a droplet collides with the plate it is assumed that it sticks on the surface et does not re-evaporate under the effect for example of the air flow. We analyzed the air flow velocity and the flow static pressure (Pa), relative to the ambient pressure to the collection system (i.e., plate) (Figures below). As pointed out by the reviewer, it can be noted that there is a zone close to the surface of the plates where the velocity is very reduced, possibly resulting in a zone of air stagnation close to the surface of the plate, which could prevent the droplets from being carried towards the edge of the plate. On the other hand, it can be noted that on the edges of the plate, the air flow, even if it is reduced compared to the velocity at the level of the inlet of the collector, could potentially induce the droplets to be blown out or even evaporated from the surface. This could be one of the explanations of the higher cut-off diameter we estimate for our collector. But this phenomenon is also surely the same for other cloud collectors where the air flow goes around the surface of collection.

[Figure]

5- There are a number of issues about the use and description of the CASCC2 that need to be clarified in a revised manuscript:

- The normal CASCC2 as described by Demoz et al. has a polycarbonate body, Teflon collection strands, and a Teflon collection trough. Was the CASCC2 borrowed for this work modified to metal construction?

Yes, it has a metal construction. The modified CASCC2 from IRSN has a metal body, stainless-steel collection strands, and a metal collection trough. This is now indicated in the manuscript: "This collector has a metal body, stainless-steel collection strands, and a metal collection trough." (section 2.3.2).

- The sampling performance of the CASCC2 depends on the fan used to pull the airflow. The analysis in Demoz et al. is based on a 115VAC Nidec-Torin TA700 fan operated at 60 Hz. What fan was used in the CASCC2 operated in the current study? The fan performance will change even with the 220 V/50Hz version of the TA700 fan. In particular, the flow rate will decrease with a lower frequency voltage supply. This should be considered in the performance analyses.

The fan that is used for the modified CASCC2 is an "4100 NH5 - S-Force ebm-papst, 24 V dc, 119 x 119 x 38mm, 45W" that was operating a frequency of 60 Hz allowing to induce an air velocity in front of strands equal to 8.6 m s$^{-1}$ that has been measured by a thermal anemometer at different levels. With the surface area of the collector inlet, we were able to estimate the air flow entering the collector at 348 m$^3$ h$^{-1}$. In Demoz et al. (1996), two factors have been proposed to estimate the fraction of the air that actually induced the sampling of the droplets: the volume fraction of the air sampled (86%) and the volume fraction of the collected ambient droplet distribution (95%). This leads to a final sampled air flow of 284 m$^3$ h$^{-1}$. We modified the text to explain this in a more descriptive way (section 2.3.2).

- The CASCC2 is sometimes operated with a downward facing inlet excluding rain. Was that used here?
No, the CASCC2 used in the study was not operated with a downward facing inlet to exclude the collection of rain.

6- Lines 357-360: The authors make a critical error here in the calculation of the BOOGIE air sampling rate. They should not assume that the exit velocity matches the velocity through the impactor slots. Rather, they should use conservation of mass and assume that the exit volumetric flow rate matches the total volumetric flow rate entering the collector, adjusting for the pressure drop through the collector. Their incorrect assumption of matching velocities introduces a critical error into the collector sampling flow rate that will bias many of their subsequent analyses. This error needs to be fixed, sample flow rate corrected, and derived LWC and collection efficiency also corrected. This is a critical error and must be fixed!!

Yes, we totally agree with this. See responses to the questions n° 2 and 3 where we described the corrections we performed regarding this error. Sample flow rate was corrected as well as the derived LWC and the collection efficiency.

7- The same error discussed in item 6 appears to have been made for considering the CWS flow rate requiring similar correction and updating of results.
Yes, we totally agree. This has been corrected following your comment. We recalculated the collection efficiency based on the corrected airflow.

8- The authors should introduce and define the concepts of isokinetic and non-isokinetic sampling much earlier in the paper and address them when discussing collector performance. They should also be clearer in discussing when and how the BOOGIE collector was aligned with respect to the wind in the collection efficiency evaluation.
Yes, we totally agree with your comment.
First, the collector was installed in front of the wind at the beginning of the experiment. For this we used the wind direction data that is available at the station. During the sampling we regularly checked the wind speed direction, and we modified the orientation of the sampler accordingly.

Regarding the concepts of isokinetic and non-isokinetic sampling, this is related to the wind intensity that can modify this condition. For safety reasons, of course, we avoid installing the collectors when the wind speed is too high (more than 50 km h$^{-1}$), and also because of this problem of non-isokineticity during the sampling.

In Table S3 you will find the information relative to the wind speed measured during the sampling of the 21 studied cloud events. The average values during the duration of sampling can vary from 0.2 to 14.3 m s$^{-1}$ (0.7 to 50 km h$^{-1}$). Only the clouds collected during the 02/07/2016, the 25/04/2022, the 03/04/2024 and the 15/04/2024 present value above 8 m s$^{-1}$, respectively at 12 $\pm$ 1.4 m s$^{-1}$, 11 $\pm$ 1.2 m s$^{-1}$, 13.9 $\pm$ 1.3 m s$^{-1}$ and 14.3 $\pm$ 1.4 m s$^{-1}$. Those values are closed or higher to the air flow velocity of the sampler (14 m s$^{-1}$). For these 4 events, the collection of droplets has been performed under possibly non isokinetic conditions. A problem with the orientation of the collector in strong wind condition can lead to significant gaps in collection efficiency. We cannot rule out the possibility that at some point the collector may not face the wind, leading to a reduction in collection efficiency, or that it may face the wind at very high intensities, leading to sampling in non-isokinetic conditions and inducing collection efficiencies more than 100%. This is clearly seen in these 4 events, which show highly heterogeneous collection efficiencies. A discussion on this point is implemented in the new version of the manuscript (section 3.3.2).

9- Figure 3. The authors should add drop sizes to the legend. In addition to the model results in Figure 3, the authors should present modeled collection efficiency (e.g., at 10 m/s) vs. drop size. This is how collector performance is typically shown. Finally, the authors should use jet velocities rather than exit velocities as the independent variable in Figure 3.

Yes, we agree that the drop size is missing on the Figure. This is not indicated in Figure 3. For the Figure 3, we prefer to present the result for different outlet velocities because we measured it experimentally and the air inlet velocity is heterogenous at the slot level. We apply the constrain on the air outlet velocities to perform the CFD simulation.

10- The authors should more clearly point out that drop composition within a cloud has been experimentally shown to vary with drop size (e.g., Bator and Collett (1997) Cloud chemistry varies with drop size. J. Geophys. Res., 102 (D23), 28071-28078) and that differences in the composition of collected samples may therefore be expected if the lower cut size for the collector changes. They kind of dance around this concept but never explain it clearly.

Yes, of course, we completely agree that the cloud composition is variable and droplet size dependent. Most of the chemical compounds measured in this study were close between the three samplers. For those which present different concentration, it could be partly due to the difference of size cut-off diameter between the samplers. We have already mentioned this point in the manuscript as one of the causes of this composition difference :

"These three samplers present specific designs and surfaces of collection (plate for BOOGIE and CWS vs strands for CASCC2), leading to different estimated cutoff diameters (around 12 μm for BOOGIE, 7.5 μm for CWS, and 3.5 μm for CASCC2) and possibly to small differences in the chemical composition of the samples.".

However, the real cut-off diameter for the 3 samplers and more generally for all samplers are only estimation; moreover, for the BOOGIE collector, we use CFD simulation to estimate the cut-off diameter that has not been performed in the same way for other samplers. For this reason, we think it is more prudent to not only explain the difference of chemical composition with the difference of in the cut-off diameters.

This issue about the size cut-off and the chemical composition is a major item for our scientific community. The idea of the present paper was not to make an in-depth comparison of the measurements obtained between different collectors but to mention that the measurements obtained were consistent with each other. The main objective of the paper is to present the collector and its efficiency. Moreover, within the framework of the European ACTRIS network, a special effort is being made on these aspects: intercomparison campaigns between collectors are being conducted (in which BOOGIE has been deployed) to compare their collection efficiency, and intercomparisons between cloud chemistry measurements are also being carried out. This work is still in progress, but initial results confirm that our collector is efficient, and indicate that laboratory measurements (mainly ions) are comparable between collectors, even if some bias may appear. This work will be promoted within ACTRIS (see The Centre for Cloud Water Chemistry (CCWaC): https://www.actris.eu/topical-centre/cis/centre-cloud-water-chemistry-ccwac).

11- Figure 5: I wonder if these comparisons would be better represented by scatter plots of concentrations measured in the collector pairs. As presented, it is unclear if larger differences might simply reflect measurements of species at very low concentrations.

For the most concentrate ions (*i.e.*, the major ones contributing to the Totat Inorganic Content - TIC), they are comparable as indicated in the paper for sulphate, nitrate, chloride, and ammonium. The compounds presenting the higher variability are the less concentrate ions such as magnesium (<15 µM) and potassium (<8 µM).

This sentence have been added in section 3.4 :
"A large variability of a factor 3 to 6 was observed for magnesium and potassium ions, but they also had the lowest concentrations, below 15 and 8 µM, respectively (Figure S12). For the most concentrated ions, such as ammonium (over 150 µM) and nitrate (over 50 µM), their concentrations are comparable between samplers.".

Here, we compared the difference between BOOGIE and CWS, and BOOGIE and CASCC2. The variability between CWS and CASCC2 is also high for the low concentrate ions.
By using the discrepancy factor (Df), we have probably artificially brought to the fore differences that are not the most significant because they result from the least concentrated ions. Looking at the Figure S12 in the supplementary information, these differences do not appear as pronounced.

[Figure]

Figure S12. Histograms presenting the concentrations of anions and cations for the three cloud samples collected using CWS, BOOGIE, and CASCC2 in parallel.

12- The authors occasionally allude to BOOGIE being capable of supercooled cloud sampling. Has this been evaluated? Are the impaction surfaces easily removable to retrieve accumulated rime? Are there

issues with the collector face or jet entrances riming and/or clogging? What about the fan? With drops up to 10 um transiting the fan I would think this could be an issue.

[Figure]

The BOOGIE collector could be deployed under supercooled condition. The impaction plates are easily recovered after sampling and the ice formed on the plates can be collected in bottles (see photo).

The collection period must be short to avoid clogging the slots and causing fan failure. This type of sampling can therefore be carried out, but we are not very confident about the measurements that will be made on this sample, particularly concerning volatile species, possibly not trapped in the ice formed. We avoid collecting supercooled cloud event, and even if we analyzed the sample chemical and biological composition from these events, these data were never considered in our data base.

In addition, the geometry of the collection surface changes over time, leading to a bias in drop collection efficiency. Consequently, we sincerely prefer to avoid collecting under supercooled conditions for all these reasons. Regarding the reviewer comment, we do not find in the manuscript where we mentioned the possibility to collect with BOOGIE under supercooled condition. We only mention that the CWS can operate under this condition and that it exists an upgraded version of the CASCC2 was designed for supercooled cloud sampling: the Caltech Heated Rod Cloud Collector (CHRCC). We prefer to not add additional text relative to the possible sampling of supercooled cloud with the BOOGIE collector, to avoid any confusion.

Minor items:

We would like to thank the reviewer for taking the time to look carefully at the small errors in the manuscript. We answer below.

- The manuscript does not adequately describe turbulent conditions in the collector. As far as I can tell, the only evaluation of turbulence is through the CFD modeling and that is not well described. There are also ambiguous claims in the abstract and conclusions about "few turbulences have been observed…". Were there any "observations" of turbulence and, if so, what do these claims mean?
Yes, we agree with the reviewer comment. Our conclusions are not effectively based on a turbulence calculation/assessment in the collector. Turbulence is considered in numerical CFD simulations, as mentioned in section 2.2. Of course, some amount of turbulence could certainly be observed, particularly around the collection zone. We have therefore chosen to delete these sentences to avoid drawing overly hasty conclusions.

- Line 103: change "where and their designs" to "where their designs"
Done.

- Line 107: Change "Caltech University" to California Institute of Technology"
Done.

- Lines 112-113: Her the authors refer to "The sampler". This is confusing as written. I first thought this was referring to the BOOGIE sampler but I think it actually refers to the CWS. Please rewrite this sentence to be clear.
Thanks, we rewrite this sentence following your comment.

- Lines 198-199: please specify here the optical instrument used to measure LWC (PVM-100).
Done.

- Line 223: replace "resistance" with "drag"
Done.

- Line 228: replace "strains" with "strands"
Done.

- The authors should define the drop effective radius and clearly point out that this is not the mean physical radius.
Line 353: we clarify this point following your comment.

- Lines 396 and 397: Change "PWM" to "PVM." Why can't the PVM output be recorded at higher time resolution? It's just a voltage.
Done. For the time resolution, since this instrument is used to monitor the LWC over long period of time, the data acquisition frequency is 5 min for data storage reasons.

- Line 435: The sentence here is garbled. Please rewrite.
Done.

- Line 531: Replace "in a radius" with "in radius"
Done.

- Line 548: Replace "would present" with "would likely present"
Done.

- Line 560: It would be worth pointing out that there is a version of the CASCC family specifically designed for supercooled cloud sampling: the Caltech Heated Rod Cloud Collector (CHRCC). This is discussed by Demoz et al. and other references.
We add this information in the description of the CASCC2 (section 2.3.2).

---

## Author Comment (AC3)

**Reviewer 3**

Vaitilingom et al. present the design of a new bulk cloud water collector (BOOGIE) which has been tested and applied at the Puy de Dome station and will be used for studies of biological and chemical composition. The new collector is well-motivated by shortcomings of existing designs especially with regards to sampling rates and ease of operation for biological applications and its characteristics and comparisons with other designs are reported and discussed. I have a number of comments and suggestions the authors might want to consider, and I recommend publication after these have been appropriately addressed.

We would like to thank the reviewer for his review and the high regard in which it holds the development work that was carried out. We answer to his/her comment below in blue.

Major comments

1- The intercomparison of solute concentrations (inorganic ions, formaldehyde and H2O2) between different collector designs is at the core of the collector evaluation, because this is what these instruments are made for in the first place. Unfortunately, this part of the manuscript is a bit weak. Only two samples were taken in parallel with the CWS, which has been in operation for more than 20 years at the station, and three with the CASCC2, another popular design. Based on such low statistics, it is difficult to robustly judge on the comparability of cloud water solute concentrations. I wonder these are really the only available data to compare? Given that the first BOOGIE application took place 8 years ago already, I would have expected a larger database for comparison. In case there is more data, I would suggest to include it to the manuscript, which would much strengthen the intercomparison section.

We agree with you comment of course. Unfortunately, we did this intercomparison in 2016 when the BOOGIE was under evaluation. We did not perform other inter-comparisons between the samplers at the puy de Dôme station. There are several reasons for that. The first most important point is that these intercomparisons of cloud collectors is currently underway in the frame of the European ACTRIS network. Measurement campaigns at various sites (Mont Schmücke, Germany; Puy de Dôme, France; Mont Sonnblick, Austria) have been carried out over the course of 2023 and 2024 to intercomparison the collectors deployed on a European scale: their collection efficiency under various environmental conditions and the resulting chemical water composition. This work is currently being finalized, and without revealing too many conclusions about it, it highlights the good performance of our collector. The second point is that we do not own the CASCC2 collector since it belongs to our colleagues from the IRSN institute, in France; so we could have only performed intercomparisons between the BOOGIE and CWS collector that is not sufficient to our point of view. Finally, since the BOOGIE collector presented high collection rates, over the last few years, we have prioritized the use of the BOOGIE collector vs the CWS to sample clouds at the puy de Dôme. The section in the paper on chemical composition intercomparison is admittedly rather preliminary and will in future be conducted within the ACTRIS network. We hope these arguments convince you.

To strengthen the section on the efficiency of collection, we added 4 cloud events (i.e., 19 samples) to the studied database.

2- The intercomparison of ion concentrations is discussed in a bit of an optimistic way. Deviations of up to a factor of about 2 (Df < 0.5) are assessed as "good comparability", but it remains unclear what this positive assessment is based on. Would systematic deviations in this range not pose a problem to the long-term trends of concentration data at the station? How do the deviations compare to earlier collector intercomparison exercises, for individual ions as well as overall? In a few extreme cases, deviations up to factors of 3 and 6 were observed between BOOGIE and CASCC2, but these are hardly discussed.

Instead, it is implied that CASSC2 concentrations "appear" slightly higher" and only "at first glance". I'd recommend to revise this section in a more detailed and more critical way. Overall, the results seem to call for more systematic intercomparisons with a larger number of samples.

The intercomparison of ion concentrations is a delicate point to interpret. Globally, most of the chemical compounds measured in this study were similar from the three samplers. It is the case for formaldehyde, hydrogen peroxide and some inorganic ions, as well for the microbial parameters (ATP/ADP). For the most concentrate ions (*i.e.*, the major ones contributing to the Totat Inorganic Content - TIC), they are comparable as indicated in the paper for sulphate, nitrate, chloride, and ammonium. The compounds presenting the higher variability are the less concentrate ions such as magnesium (<15 µM) and potassium (<8 µM).

This sentence have been added :

"A large variability of a factor 3 to 6 was observed for magnesium and potassium ions, but they also had the lowest concentrations, below 15 and 8 µM, respectively (Figure S12). For the most concentrated ions, such as ammonium (over 150 µM) and nitrate (over 50 µM), their concentrations are comparable between samplers.".

Here, we compared the difference between BOOGIE and CWS, and BOOGIE and CASCC2. The variability between CWS and CASCC2 is also high for the low concentrate ions.

By using the discrepancy factor (Df), we have probably artificially brought to the fore differences that are not the most significant because they result from the least concentrated ions. Looking at the Figure S12 in the supplementary information, these differences do not appear as pronounced.

[Figure]

Figure S12. Histograms presenting the concentrations of anions and cations for the three cloud samples collected using CWS, BOOGIE, and CASCC2 in parallel.

Following the reviewer's comment, we tested new figures to highlight the differences between collectors. For example, we proposed the radar plot but it seems not really adequate as it is difficult to read.

[Figure]

Radar plot describing the differences between the collectors for the 3 cloud events.

We prefer the histograms proposed in the SI (Figure S12).

We used to see variable concentration even for a same cloud event with identical collector. An example was given in the SI text, see Fig S13.

[Figure]

Figure S13. Histograms presenting the concentrations for a specific cloud sampled on 08/07/2021 at PUY with two BOOGIE collectors. This time, three aliquots were analysed twice (error bars) using ion chromatography. p-values are indicated with the black line and the yellow dashed line indicates the threshold of $p = 0.05$.

More generally, inherently linked to the collection method, the composition of the water sample is not a perfect representation of the real cloud water composition. Through the impaction process by all passive and active samplers, a bias is added. Several parameters can affect the cloud composition: the collection temperature, the matter of the body collector and the bottle, the time delay between the impaction and its flowing inside the vessel, the variation of air/liquid volume of the collection bottle. These conditions affect the liquid/gas exchange, even the microorganisms will modify the chemical composition by their metabolic activity during the sampling time. For all these reasons, we consider the difference of composition in our study acceptable even for the less concentrate compound.

But as mentioned by the reviewer for a long-term trends of concentration data at the station, a full comparison between the cloud samplers is required. We work on it, a special effort is being made on these aspects at the European level: intercomparison campaigns between collectors are being conducted

(in which BOOGIE has been deployed) to compare their collection efficiency, and intercomparisons between cloud chemistry measurements are also being carried out. This work is still in progress, but initial results confirm that our collector is efficient and indicate that laboratory measurements (mainly ions) are comparable between collectors, even if some bias may appear. This work will be promoted within ACTRIS (see The Centre for Cloud Water Chemistry (CCWaC): https://www.actris.eu/topical-centre/cis/centre-cloud-water-chemistry-ccwac).

3- There are a number of inconsistencies for some quantitative results between abstract, main part and conclusions, which I will give below. In general, I'd suggest to critically check that correct and consistent information is given everywhere throughout the manuscript. Also, the wording and sentence structure would benefit from a critical check here and there.

Yes, we totally agree with your comment. We carefully check this in the revised version of the manuscript. We hope this is now correct. Concerning the wording and sentence structure, before the submission, we opted and paid for a high-standard language editing service to check the writing. We have already made this effort, so if the reviewer ask for additional corrections, we think that the journal might be able to take on this task.

4- Fig. 4 and its discussion are a bit misleading. Based on the data points, I believe the authors have forced their regression through zero, which means their slope, intercept and coefficient of determination are incorrect. Forcing through zero should only be done for very good reasons. In this case, the intercept could actually be informative and should not be artificially removed. With a simple linear regression, I would expect a slightly negative intercept and a slope close to 1. Given that small unrecovered amounts of water from inner surfaces or even slight evaporation inside the collector seem plausible, I would not expect an intercept of 0. Also, the collection efficiency does not seem to decrease at higher LWC, based on just the data points. This seems to be an artifact from the applied regression through zero. Please carefully check and revise accordingly.

Yes, we forced the regression through zero. Following the reviewer comment regarding the possibility of some artefacts (small unrecovered amount of water from inner surface or possibly slight evaporation inside the collector that we cannot avoid), we changed the regression. The intercept is now equal to -0.02, a negative value as envisaged by the reviewer and the slope equal to 0.92.
Concerning the relationships we described (decrease of collection efficiency at higher LWCmes), this is true, there is no clear tendency. We changed the text accordingly.

5- L357ff: From the explanations, it seems outlet velocity (not inlet velocity) together with inlet surface area has been used to calculate the air volume flow rate. This would be wrong and would need to be corrected throughout the paper, including any discussions and conclusions that might change with a corrected air flow rate.

The inlet velocity is estimated by measuring the outer flow. Understanding the questioning of the reviewers, we have experimentally evaluated new flow rate measurement during the revision time delay, and this is presented in a new section (3.2 Evaluation of the air flow inside the BOOGIE collector). Consequently, with a new air flow, we recalculated the collection efficiency of the BOOGIE collector and modified the discussion accordingly.

We added in the manuscript the following paragraph in the new section "3.2 Evaluation of the air flow inside the BOOGIE collector" describing the experiment we conducted to estimate the sampler air flow: " [...] Therefore, we designed an experiment to measure the air flow at the collector outlet. The airflow rate at the fan outlet was measured using the following procedure. A 3.5 m long PVC pipe with an internal diameter of 154 mm was installed after the fan outlet. This diameter enables the entire flow to

be measured without reduction, thus limiting the additional pressure losses generated by the addition of the pipe. A hot-wire anemometer was installed in the tube at 3 m from the fan. The large distance/diameter ratio (greater than 19) minimizes disturbances (high turbulence and vortex rates) as the air passes through the axial fan.

The flow velocity profile is measured every 5 mm along the diameter. Flow rate is calculated by summing the average velocity for each ring by the ring area. The flow rate was estimated at 433 $m^3$ $h^{-1}$ at 90% of the fan speed. The average velocity in the pipe is found by dividing the flow rate by the cross-sectional area, which corresponds to a velocity of 6.5 m $s^{-1}$. Based on this velocity, the Darcy-Weisbach formula and the Moody diagram (with a relative roughness of 2 $10^{-5}$), the pressure drop in the pipe is estimated at 10 Pa. As a result, the addition of the pipe has little influence on the flow rate.

The pressure drop in the BOOGIE impactor can be estimated from the fan and flow characteristics. Since the flow rate has been calculated at 433 $m^3$ $h^{-1}$, the pressure drop compensated by the fan is estimated at 220 Pa, and consequently the pressure drop in the impactor is around 210 Pa. The variation in density is less than 0.0025 kg $m^{-3}$, i.e. a variation of less than 0.25%. The flow can be considered incompressible, and conservation of flow-volume can be used. The average velocity at the BOOGIE inlet is estimated at 11 m $s^{-1}$, by dividing the flow by the inlet cross-section of 10.9 $10^{-3}$ $m^2$. This average velocity differs from the measured velocity at inlet (14 m $s^{-1}$) due to the velocity profile at the slots. The measurement corresponds to a maximum velocity.".

In the previous version of the manuscript, based only on the air inlet measurements, we did calculations that were false as pointed by the reviewer due to the non-respect of the conservation of mass. Note that, in the CFD calculation, of course we do not violate the principle of conservation of mass.

We therefore re-calculate all the collected LWC ($CLWC_{exp}$) and we redid the figure 4 that present experimental collected LWC vs the measured LWC. The collection efficiency (in %) were also recalculated and are indicated in Table S3 in the Supplementary Information. We also decided to include in this study additional clouds collected at the puy de Dôme, notably in spring 2024 during an international RACLET measurement field campaign as part of the ATMO-ACCESS program. 4 cloud events corresponding to 19 samples were added to our analysis.

Further issues

- The title should be revised as it is uncommon to include two colons. Also, it becomes clear at second glance only, why oxidants and metal elements are mentioned. These are not measured and discussed in the manuscript, but seem to be part of the collector name. This could be made clearer, e.g. like this: "Design and evaluation of a new cloud water collector for the analysis of biological, …, and metal elements (BOOGIE). Actually, nowhere in the paper it is explained where the name BOOGIE originates from.

We simplified the title of the paper following your comment. Now, the title is shorter: "Design and evaluation of BOOGIE: a collector for the analysis of cloud composition and processes". At the end of the introduction, we explained why we decided to name the collector BOOGIE. This refers to the all the elements that can measured after the collection of the water: "Biological, Organics, Oxidants, soluble Gases, inorganic Ions and metal Elements".

To be honest, the collector's name "boogie" is a tribute to an "entity" that has been important in cloud-related activities on the Puy de Dôme site. Of course, linking this acronym to the description of what the collector does can be questionable. Our aim was not to oversell what our collector can do, since it only collects water! Collecting cloud waters among various environmental conditions is only done with the objective to perform analysis in the lab helps that are crucial to assess the effect of clouds on atmospheric chemistry.

- Results of the collector intercomparison should be summarized in a more quantitative way in the abstract and conclusion sections. The purely qualitative and subjective statement of "comparable" concentrations is not very informative. To not lengthen the abstract further, the introduction on the importance of cloud water composition could be shortened.

We modified the abstract following the reviewer comment: we shortened the introduction of the abstract and added results from the collector intercomparison. We also suppress in the conclusion the paragraph describing the different collectors used for the intercomparison experiment because it is redundant. We prefer not to add more details in the conclusion regarding the collector intercomparison. We made this choice to avoid redundancy.

- L28: Doe you mean "lightweight" or "simple" or "easy to operate" mobile sampler?

Yes, this sentence is unclear. Following your comment, we added "easy to operate".

- L33 and L346: The mean collection rate is inconsistent between abstract and results section. Please check and correct.

Yes, this is true. The good collection rate is $100 \pm 53$ mL h$^{-1}$. This has been corrected.

- L60: What does HAP mean? Make sure to introduce all abbreviations at first occurrence.

We are sorry, HAP is the French abbreviations. We modified by "polycyclic aromatic hydrocarbon (PAH)".

- L103: delete first "and"

Done.

- L112: Was the sampler indeed obtained from Kruisz et al. or built according to their design?

We modified this following your comment.

- L256: Do you mean "More information on this analysis is given in …"?

We modified the text following your comment.

- L306: When you say "classes of particles", do you mean just different sizes? Or has anything else been changed for the different "classes"?

Yes, you are right. We injected droplets from different sizes of droplet in the collector. We modified the manuscript to avoid any confusion.

- L310-311: Incomprehensible sentence, please rephrase.

This has been modified, thanks.

- Fig. 3 would be easier to understand, if droplet sizes were given directly in the legend, rather than arbitrary "class" numbers. Also, I'd suggest a more compact y-axis label without uncommon abbreviations, e.g. "number collection efficiency" and "mass collection efficiency"

Yes, we have modified the figure following your comment.

- L315ff: It is not always clear which type of collection efficiency is discussed (number vs. mass). Please make sure to correct accordingly.

This is true. We modified the text accordingly.

- L316: At velocities below 5 m/s, the collection efficiencies of large droplets are not always > 50%. Please check and revise sentence.

Yes, this is true. This has been corrected following the comment.

- L319: Does the given average collection efficiency not strongly depend on the droplet sizes which were simulated? If more larger sizes were simulated, the average collection efficiency would be different. Please clarify what can be learned from this calculated average efficiency?

The average collection efficiency depends on the simulated droplet sizes. We have chosen the different sizes based on the size distribution encountered in clouds. The largest droplets present a diameter of 20 µm. Above all, the average efficiency shows the importance of the largest droplets, both in mass and number, in the total collection.

Of course, if we add supplementary droplet size the average collection efficiency should be modified. This change would be most noticeable in term of mass collection efficiency, as the largest droplets would make a significant contribution to the total mass. But this simulation performed were only done to evaluate which droplet sizes are the most efficiently collected. The average collection efficiency is shown in the figure for information only, from our point of view.

- L324-328: This section is difficult to understand and partly redundant (50% cut-off at 10 microns). Please revise.

Yes, this is true. We modified the text following this comment.

- L335: Why are the theoretical efficiencies assessed as "good"? Are there any comparisons with other data that would substantiate this judgment?

Yes, these conclusions are not supported by quantitative data, such as comparisons with other collectors. Therefore, we modified the sentence as following:

"However, the performed simulations indicate that the new BOOGIE collector is able to collect cloud droplets, which also confirms that the distance between the air inlet slots, and the outlet fan is adequate because it is beneficial for air flow stabilisation."

- L340ff: Please check for repeated, i.e. redundant information.

This has been modified accordingly.

- L342: It would help the reader to give a brief summary of the main characteristics of the sampled events.

This has been added in the manuscript following this comment.

- L354: In the section before, the optimal velocity is given as 10, not 8 m/s. Please check and correct.

This was modified in the new version of the manuscript.

This point has been removed; a new section has been added to estimate the flow measurement « 3.2 Evaluation of the air flow inside the BOOGIE collector ».

Now the new sentence is : "To evaluate $CLWC_{exp}$, we estimated in section 3.2, the sampled air flow experimentally at 433 $m^3$ $h^{-1}$ (7.22 $m^3$ $min^{-1}$)."

- L359: In L164, the total inlet surface is given as 11.088E-3, please check and correct.

This has been changed in the new manuscript.

- L373 and elsewhere: LWCmeas, not LWCmes

This has been modified.

- L396: PVM-100

Done.

- L397: The acquisition rate of the PVM could easily be increased, but I doubt it is really a relevant factor here.

We used the LWC measured by the PVM to estimate LWCmeas that allows calculating the sampling efficiency of the collector. Recording the LWC only every 5 minutes may lead to errors in the estimation of the real LWC during sampling. Indeed, the LWC could vary significantly between two measurement points. Of course, this depends on the conditions encountered during the sampling. We would prefer to leave this point to our discussion of potential sources of error in assessing the efficiency of collecting.

- L398: Why is CLWCexp "intrinsically an estimate"? In the previous section it is explained as a value from measured velocities and geometries.

Yes, this is true. The sampler airflow is not an estimate since it is calculated using measurement of velocities and geometries (surface of the entry slots). We modified the text accordingly.

- L417/418: Harmonize between "radius" and "diameter"

We modified accordingly.

- Table 1: Rain intensities during the events should be included as well. The information on possible rain drop contamination is given late in the discussion only, but is quite a relevant information.

Unfortunately, rain is not measured at PUY with a pluviometer. In the case of rain during the sampling, we indicate this event in the text : "In the case of the 4$^{th}$ June cloud, the appearance of fine rain during sampling could possibly explain the higher of collection efficiency observed for all collectors, as we did not observe conditions such as strong winds that could disrupt the sampling.".
We modified the legend of the table 1 as suggested : "** Fine raining event before the end of sampling".

- L438: Not sure if it is really advisable to use collected water volumes as a metric for cloud LWC.

Yes, we agree. We delete this sentence.

- Fig. 5: What does the "analytical error" of 10% represent and how was it determined? In the Supplement, "accuracy" is mentioned, but precision or repeatability would be more appropriate metrics here. Is it really correct that this value is the same for minor ions with low concentrations and for major ions with usually larger peaks?

The uncertainty of measurement varies from 6 to 10% for these ions analysis. The analytical error is obtained by the calculation of the standard deviation on two analyses of the same sample. The values obtained are comparable to those obtained by the standard deviation from 3 replicates of calibration curve using a standard solution (Multi Ion cation and anion IC standard solution, Specpure™ (Dionex)). Even for the low concentrations of magnesium or potassium measured, these values were largely higher than the limit of quantification.

- Fig. 6 could go into the Supplement, as its discussion is brief and does not add much. Instead, Fig. S13 could be shown in the main manuscript as it presents a different aspect of the evaluation, i.e. comparability between two identical BOOGIEs. It might even deserve its own subsection.

This Figure 6 is useful and presents the concentrations of formaldehyde and hydrogen peroxide of three cloud event with the three samplers; the discussion is brief because the interpretation is straightforward. Their concentrations were similar between the samplers and this information is relevant.

Yes, as suggested the Fig S13 has been added to the main text.

- In Fig. S13, why are some of the aliquots much different from the other two of one and the same sample? Measurement precision from repeated IC analysis seems to be in the low percent range and can thus not explain these deviations. How much do these "inconsistent" aliquots statistically impact the between-collector comparison?

Due to the low concentration of cloud water constituents, we make sure to analyze the samples with the highest sensitivity for the instrument used (Thermo ICS 5000+). This goal is achieved by injecting a high volume of the sample (750 µL). On one hand, the limit of detection is below 0.1 µM for less concentrated ions, on the other hand, the high concentrated ions show peaks that are sometimes broad. In the context of a complex medium, with a baseline populated by other signals, such as those related to short chain carboxylic acids, integration can be sometimes difficult and produce sometimes a variability of the signal. This is the reason why we always inject the sample at least twice. In this plot, we reported the concentrations calculated for each single injection as a strict test for both the sampling error and the analytical error. If we consider the average of the three injections (Figure S13 in the SI that is now Figure 7 in the article) the difference between the two collectors is less visible.

[Figure]

- L487: In the Supplement, it says "duplicate" analyses, not triplicate.

The sentence in the text is: "Analyses were performed in duplicate or triplicate if the standard deviation of measurement exceeded 5%, the DL was of 0.07 µM (Vaïtilingom et al., 2013)."

The analysis is performed in duplicate, and if the standard deviation exceeds 5%, a third analysis is performed.

- L504/505: These comparisons should be made in a more detailed way, i.e. ion by ion and making sure the metric for "difference" is really identical.

A more detailed comparison of the similarity and discrepancies between different collectors needs more samples, collected in different environments, to build a more robust dataset and perform consistent statistical study. This goes beyond the scope of this article. Nevertheless, this comparison is already ongoing in the ACTRIS CIS CCWAC working group and will be published in a near future.

- L505: Are these values consistent with what is discussed earlier?

We cannot focus mainly on the ionic composition of minor ions which present high variation. We agree with that, but these differences were also measured from other studies. As the one mentioned in the manuscript (Wieprecht et al., 2005).

Even during a same cloud event, the composition varies. The particulate, the gas and the liquid composition is heterogenous inside a cloud.

From our experience and from the literature where samplers were compared, yes we still agree that the chemical composition and microbial energetic states between the samplers are close in this study. The formaldehyde and peroxide contents are similar between the three samplers; the microbial energetic states are barely identical. About the 8 ions measured, 3 of them (also the less concentrated) show a high variability and 5 of them have close concentrations. Thus, globally, we consider that the composition are "comparable" and have a "good assessments" between the samplers.

- L506ff: The comparison of biological data might warrant its own subsection. Right now, it is a brief extension on the chemical comparisons, but a bit difficult to understand without more context.

The microbial measurement here gives only the metabolic energetic state. It was done to see if the collection process was stressful or not for microorganisms. The energetic state reveals a good level of microbial "health" with identical values from each collector, which indicate that the collection is not stressful from the three samplers. This information is valuable but does not deserve more discussion in this section.

- The conclusion section repeats much basic information and explanations from earlier sections. It could be improved by summarizing in a more compact way the core results and conclusions.

We agree with this comment and have modified this section. A paragraph has been deleted that presented the different collectors for example, because it was redundant.

- L559: The results presented before do not seem to support the statement of high collection efficiency of the CASSC2, it was actually the lowest one among the compared designs.

All the calculations regarding the collection efficiencies of the collectors have been updated following the reviewer's comments. The CASCC2 present now the highest collection efficiency with the BOOGIE collector.

- L560: How is the applied CASSC2 model not affected by rain? The data of the cloud event with slight rain do not seem to support this statement.

The sentence was not complete from our first draft. The original sentence was: "This active sampler is a compact version of the CASCC, in which droplets are collected by impaction on a set of six rows of stainless-steel strings; it is highly efficient in terms of collection and is not affected by raindrops owing to its design by the using of a rain shield."

It has been modified, but this paragraph summarizing the three collectors was effectively too long and redundant to the introduction. We agree with the reviewers 2 and 3 and this paragraph has been deleted. We add some precision about the CASSC2 in the M&M section :

"The collector body was stainless steel, the inlet contained the impaction rows, and the sample drainage was removed before each sampling for cleaning and sterilisation. A sterilised amber glass bottle was placed under the sample drainage during collection. The CASCC2 was also not operated with a downward facing inlet allowing to exclude the collection of rain. This cloud collector was not adapted for temperatures <0 °C because droplets freeze upon impaction on metallic strands. Note that an upgraded version of the CASCC family was specifically designed for supercooled cloud sampling, the Caltech Heated Rod Cloud Collector (CHRCC)."